# Interferon inducible X-linked gene *CXorf21* may contribute to sexual dimorphism in Systemic Lupus Erythematosus

Christopher A. Odhams[1,5,10], Amy L. Roberts[1,2,10], Susan K. Vester [1], Carolina S.T. Duarte[1],
Charlie T. Beales [1], Alexander J. Clarke [3], Sonja Lindinger[1,4,6], Samuel J. Daffern [1,7], Antonino Zito [2],
Lingyan Chen [1,8], Leonardo L. Jones[1], Lora Boteva[1,9], David L. Morris [1], Kerrin S. Small [2],
Michelle M.A. Fernando[1], Deborah S. Cunninghame Graham[1] & Timothy J. Vyse [1]

Systemic lupus erythematosus (SLE) is an autoimmune disease, characterised by increased expression of type I interferon (IFN)-regulated genes and a striking sex imbalance towards females. Through combined genetic, in silico, in vitro, and ex vivo approaches, we define *CXorf21*, a gene of hitherto unknown function, which escapes X-chromosome inactivation, as a candidate underlying the Xp21.2 SLE association. We demonstrate that *CXorf21* is an IFN-response gene and that the sexual dimorphism in expression is magnified by immunological challenge. Fine-mapping reveals a single haplotype as a potential causal cis-eQTL for *CXorf21*. We propose that expression is amplified through modification of promoter and 3′-UTR chromatin interactions. Finally, we show that the CXORF21 protein colocalises with TLR7, a pathway implicated in SLE pathogenesis. Our study reveals modulation in gene expression affected by the combination of two hallmarks of SLE: *CXorf21* expression increases in a both an IFN-inducible and sex-specific manner.

[1] Department of Medical & Molecular Genetics, King's College London, London SE1 9RT, UK. [2] Department of Twin Research & Genetic Epidemiology, King's College London, London SE1 7EH, UK. [3] Kennedy Institute of Rheumatology, University of Oxford, Oxford OX3 7FY, UK. [4] University of Applied Sciences - FH Campus Wien, Favoritenstrasse 226, 1100 Wien, Austria. [5] Present address: Genomics England, Queen Mary University of London, Dawson Hall, London EC1M 6BQ, UK. [6] Present address: Institute of Biophysics, Johannes Kepler University Linz, Gruberstrasse 40, 4020 Linz, Austria. [7] Present address: Department of Genetics, University of Cambridge, Downing Street, Cambridge CB2 3EH, UK. [8] Present address: MRC/BHF Cardiovascular Epidemiology Unit, University of Cambridge, Cambridge CB1 8RN, UK. [9] Present address: MRC Human Genetics Unit MRC IGMM, University of Edinburgh Western General Hospital, Edinburgh EH4 2XU, UK. [10] These authors contributed equally: Christopher A. Odhams, Amy L. Roberts. Correspondence and requests for materials should be addressed to T.J.V. (email: timothy.vyse@kcl.ac.uk)

Females have a clear immunological advantage over males, with reduced susceptibility to infection at an early age and a superior ability to produce antibodies and serum IgM following immune challenge[1,2]. The immunological gain in females is thought to contribute to the striking sexual dimorphism observed in human autoimmune disease – where over 80% of sufferers are female[3] – and corroborates the hypothesis that genetic risk to autoimmunity is an evolutionary consequence of positive selection for favourable immune responses to infection[4]. Systemic lupus erythematosus (SLE), an autoimmune disease characterised by anti-nuclear autoantibodies and a type I interferon (IFN) signature, displays one of the most striking female-biased imbalances (9:1) in disease prevalence. Although the underlying mechanism has yet to be fully elucidated, a prominent role of X chromosome dosage is supported by the karyotypic risks for SLE. Males with Klinefelter's syndrome (47, XXY) have a 14-fold increased prevalence of SLE compared to 46, XY males, which approximates to the prevalence seen in 46, XX females[5]. Furthermore, whereas 45, XO females have lower risk[6], SLE prevalence in 47, XXX females is ~2.5 times higher than in 46, XX females[7]. Indeed, mammalian X chromosomes, for which males are hemizygous, are enriched for immune-related genes[8].

X chromosome inactivation (XCI) is a unique mammalian dosage-compensation mechanism which equalises expression of X-linked genes between sexes[9]. This random process results in either the paternally or maternally inherited X chromosome becoming inactivated (Xi) through enriched heterochromatic modifications, which promotes gene silencing to leave one transcriptionally active X chromosome (Xa) in females[10]. However, an estimated 15% of X-linked genes, preferentially found on the Xp arm, escape XCI and thus display expression from both chromosomes, although typically expression is still lower from the Xi compared with Xa[11]. A further 10% of X-linked genes display variable expression from the Xi – an observation which itself is variable between both individuals and cell types, and throughout development and ageing[12]. It is these XCI-escaping genes, through their partial or complete lack of dosage compensation, that are thought to contribute to genetic sexual dimorphism and phenotypic differences in X-chromosome aneuploidies[13]. Furthermore, the relaxation of Xi silencing in female mammals includes increases in the expression of several immunity-related genes[14]. How genes that escape XCI contribute to sexually dimorphic diseases has not been thoroughly studied.

A SLE association at the Xp21.2 locus (rs887369; $P_{META} = 5.26 \times 10^{-10}$; OR = 1.15) was recently identified in a European GWAS and replication study[15]. Intriguingly, this locus is encoded outside the pseudo autosomal region (PAR) and the lead SNP (a synonymous variant) resides in the final exon of CXorf21, a protein-coding gene of unknown function. CXorf21 has been shown to escape XCI in lymphoblastoid cell lines (LCLs), and is one of only 14 X-linked genes that is differentially expressed between Klinefelter's syndrome (47, XXY) and 46, XY males in LCLs[16,17]. A recent whole-blood gene expression study also identified CXorf21 as one of seven genes upregulated in female SLE patients displaying disease flare relative to those with infection[18].

Despite the stark karyotypic risk, there remains a lack of understanding of the contribution of the X chromosome to SLE, which is a leading cause of death in females aged under 34 years of age[19]. Here we describe fine-mapping and characterisation of the association at Xp21.2 through complementary genetic, in silico, in vitro and ex vivo approaches using both existing and newly generated data (all methods are summarised as a flow chart in Supplementary Fig. 1). We demonstrate that the candidate gene, CXorf21, is an IFN-responder with both cell-type specific and sexually dimorphic expression amplified by cellular activation. Additionally, we provide evidence at the protein-level of CXORF21 co-localisation with TLR7; a gene causatively linked to SLE and which also evades XCI. Our study demonstrates IFN-inducible magnification of sexual dimorphic gene expression contributing to SLE risk.

## Results

**Genetic refinement of the Xp21.2 SLE susceptibly locus**. The source of all cohorts used within this manuscript along with the analyses performed are presented as a flow diagram in Supplementary Fig. 1. The UK10K-1000 Genomes Project Phase 3 reference panel[20] was firstly used to impute the Xp21.2 locus of the Bentham and Morris et al. SLE GWAS (10,995 individuals of European ancestry)[15]. Logistic regression revealed a synonymous variant, rs887369 (MAF = 0.24), to be the most significantly associated SNP ($P = 3.34 \times 10^{-7}$; OR = 1.43, 95% C.I = 1.23–1.66; Fig. 1a) and conditional analysis upon rs887369 showed no evidence of independent association signals (Fig. 1b). Haplotype analyses revealed that rs887369 tags a single, 1 kb haplotype block comprising five near-perfectly correlated SNPs mapping to the 3′ region of CXorf21 (Fig. 1c)– encoding a small, 301-amino acid protein of unknown function. SNPs rs2529517 (distal) and rs887369 (proximal) define the boundaries of the associated haplotype, which map downstream of the 3′-UTR of CXorf21, and to the gene's third and final exon respectively (Fig. 1c). Three of the five associated SNPs are transcribed from CXorf21, with rs887369 effecting a synonymous change (V209) and both rs2532873 and rs2710402 residing in the 3′-UTR. The remaining two SNPs, rs2429517 and rs2429518, are located in the downstream intergenic region of CXorf21. The associated haplotype is distinctly separated from neighbouring haplotypes by high recombination (D′ < 0.6, $r^2$ < 0.2) and accordingly, the risk haplotype itself represents the only observed association with SLE ($\chi^2 = 29.87$, $P = 4.63 \times 10^{-8}$; $\chi^2$ test; Fig. 1d).

CXorf21 is known to escape XCI[16]. We performed a statistical test on the association with rs887369 to see if a model that assumed the SNP was in an area that escaped inactivation fitted better than a model assuming full inactivation. A likelihood ratio test to fit both association models failed to reject the model of full inactivation ($P = 0.78$). Therefore, from our case/control data we have no evidence against the hypothesis that this association lies in an area of full inactivation. To extend these analyses, we determined the odds ratios of the risk alleles in females and males separately. We observed a higher odds ratio for females homozygous for the rs887369 [C] risk allele with respect to homozygous for [A] non-risk (OR = 1.58, 95% C.I. 1.29–1.93) compared to the males (OR = 1.46, 95% C.I. = 1.10–1.92), who are hemizygous for the risk or non-risk alleles. The higher odds ratio in females is likely to reflect a gene dosage effect secondary to some degree of loss of X inactivation.

**The risk haplotype increases expression of CXorf21 in LCLs**. As no protein-altering variants were identified through fine-mapping, we sought to establish whether the SLE risk alleles at CXorf21 colocalised with cis-eQTLs for gene transcription. Non-random inactivation of the X chromosome (skewing) and variability in the degree of silencing of gene expression from the inactivated X in females complicates the identification of X chromosome eQTLs in females. Therefore, to study cis-eQTLs at the CXorf21 locus, we employed two complementary methods of assessing the influence of the risk haplotype, tagged by rs887369, on the expression of genes within the Xp21.2 region: (1) using the hemizygosity of males to isolate the allelic effects; (2) removing females exhibiting strong evidence of extreme skewed XCI to

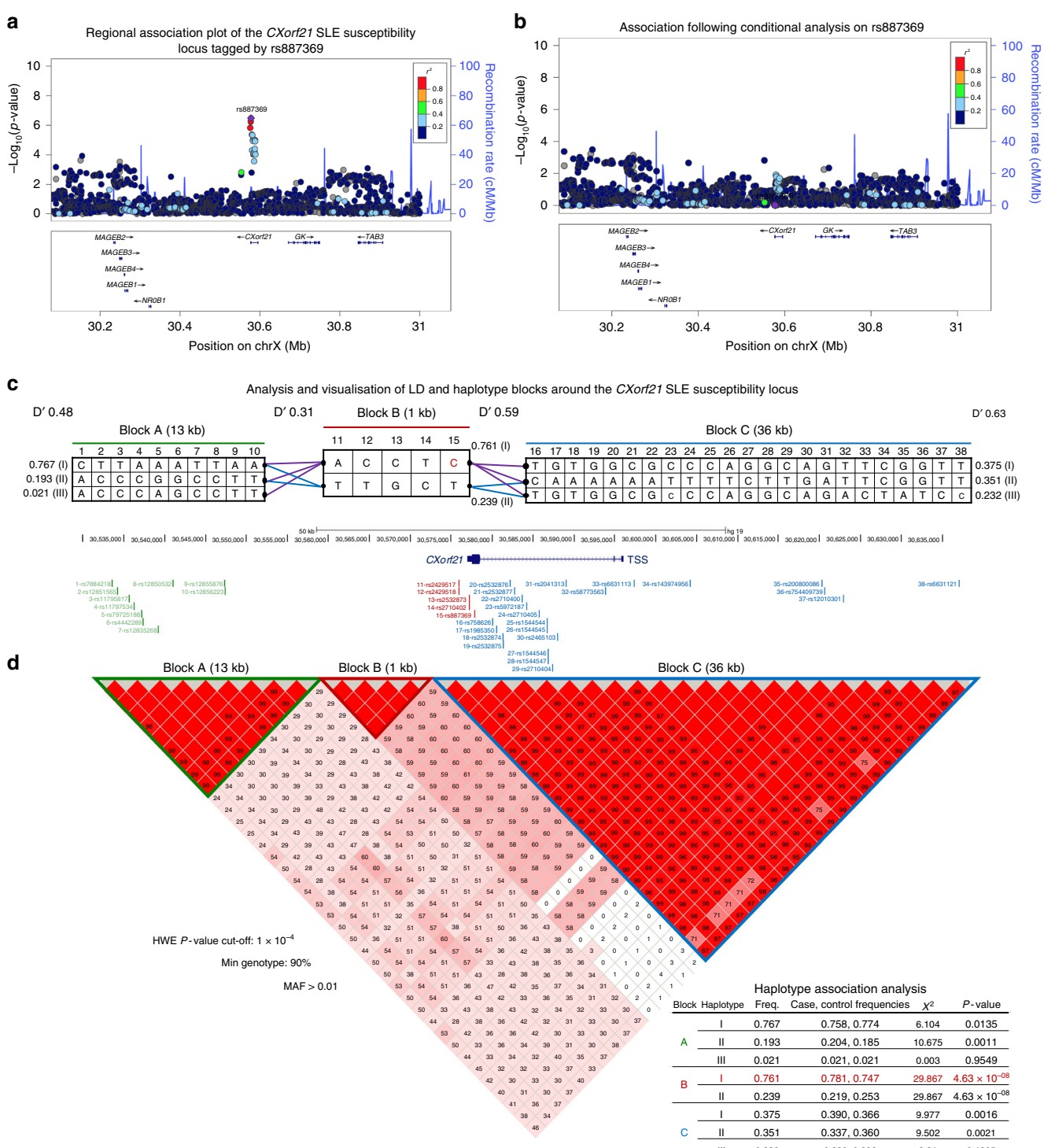

**Fig. 1** Genetic refinement of the Xp21.2 (rs887369) SLE susceptibility locus. **a** Association plot of the 1 Mb region (X:30,077,846–31,077,845) of SLE-associated region Xp21.2 following genotype imputation to the level of UK10K-1000G Phase III and association testing as described ($n = 10,995$ individuals of European ancestry). rs887369 is shown as the most significantly associated lead SNP. Genetic association plots were generated using LocusZoom. **b** Association plot of the 1 Mb region following conditional analysis on lead SNP rs887369. **c** Haplotype construction and visualisation of the Xp21.2 SLE susceptibility locus conducted in Haploview 4.2. The top panel shows the structure of the three blocks and haplotypes surrounding the lead SNP rs887369 (highlighted in red, block B, SNP #15). Blocks are separated by regions of high recombination as specified by D' and $r^2$. The frequency of each haplotype is denoted. The middle panel presents the colour-coded haplotypes and individual SNPs by their genomic coordinates around CXorf21. The bottom panel shows the LD structure and pair-wise correlation ($r^2$) of SNPs, and length of each block. **d** Right table: case–control association analysis of each haplotype using Haploview 4.2

reduce the variability in the degree of skewing of X-chromosome expression.

The associated haplotype, tagged by rs887369 [C], correlated with increased expression of *CXorf21* in LCLs from male samples in the Geuvadis RNA-Seq dataset ($\beta = 1.56$, $P = 1.94 \times 10^{-03}$; linear-regression; Fig. 2a). The expression of neighbouring genes *GK* and *TAB3* showed no significant association with rs887369 ($P = 0.7$ and $P = 0.09$, respectively, linear-regression, Fig. 2a). Many variants may act as *cis*-eQTLs, however it is important to note that rs887369 was the most significantly associated *cis*-eQTL for *CXorf21* (Fig. 2b) and the remaining *cis*-genes (the *MAGEB* family and *NR0B1*; +/−1 Mb from rs887369) were not expressed in LCLs (RPKM < 1).

The allelic effect on *CXorf21* expression was only nominally significant when performing the *cis*-eQTL analyses in female individuals from Geuvadis RNA-Seq dataset in LCLs ($P = 0.02$; linear-regression; Supplementary Fig. 2a). In order to investigate *cis*-eQTL effects at rs887369 in females, we interrogated an additional RNA-Seq gene expression dataset in LCLs constructed exclusively from female donors from the TwinsUK cohort[21]. This dataset was selected for analysis as it had been previously analysed for skewing of X chromosome inactivation using allele specific expression (ASE) of the *Xist* silencing lncRNA (manuscript in preparation). In order to study potential *cis*-eQTLs at the *CXorf21* locus, we removed individuals showing marked skewing, in whom the *Xist* ASE showed that one parental X chromosome contributed less than 20% of the *Xist* expression. In this subset of 412 non-skewed individuals, we observed a statistically significant increase of *CXorf21* expression with respect to the rs887369 [C] risk allele in females ($P = 7.00 \times 10^{-03}$; linear-regression; Fig. 2c).

We validated this effect in vitro by qPCR of independent LCL samples selected from the HapMap Project on the basis of their genotype at rs887369. In these cells a 1.9-fold increase of *CXorf21* mRNA was detected between rs887369 homozygous risk and non-risk females ($P = 4.1 \times 10^{-5}$; *t*-test ;Supplementary Fig. 2b). Following validation of the anti-CXORF21 antibody (Supplementary Fig. 3), the observed increase in expression by the risk allele was found to persist at protein-level ($\beta = 0.49$, $P = 2.88 \times 10^{-5}$; Fig. 2d; raw data are shown in Supplementary Fig. 2c).

**Risk variants increase *CXorf21* expression upon activation**. We expanded our analysis and interrogated a genotype-expression cohort from a range of human primary ex vivo immune cells. When assessing male samples only, we found that the associated haplotype, tagged by rs887369, was a significant *cis*-eQTL for *CXorf21* in both Lipopolysaccharide (LPS) stimulated ($P = 1.08 \times 10^{-03}$) and IFN-γ-stimulated ($P = 1.10 \times 10^{-3}$; linear-regression) monocytes (Fig. 2e). The [C] risk allele once again correlated with increased *CXorf21* expression. Interestingly, no statistically significant *cis*-eQTL associations were observed in the unstimulated experiments: B cells, NK cells, neutrophils and monocytes, which suggests an activation-state specificity of the *cis*-eQTL. When the same analysis was performed in the female samples of the same cohort, no significant *cis*-eQTLs were detected in any of the cell types (Supplementary Fig. 2d).

**Epigenetic fine-mapping of the Xp21.2 associated haplotype**. Using the Roadmap Epigenomes Project[22] (12 different histone marks across 127 cell and tissue types), we used chromatin fine-mapping to functionally prioritise the five SNPs carried on the 1-kb associated haplotype. The associated SNPs localised only to a single histone modification, H3K36me3, across five cell types: blood mononuclear cells, peripheral blood B cells, monocytes, neutrophils and the lymphoblastoid cell line GM12878. Analysis of the signal value distribution of H3K36me3, designating regions

of active transcription, across these cell types revealed that rs887369 localised to the binding site summit of H3K36me3 whilst the remaining four SNPs on the haplotype localised to the tails of the signal distribution (Fig. 3a). The greatest enrichment of H3K36me3 across the entire *CXorf21* gene locus was concentrated to ±100bp of rs887369 in monocytes ($P = 6.1 \times 10^{-14}$; MACS2) and neutrophils ($P = 2.0 \times 10^{-17}$; MACS2; Fig. 3b). The rs887369 SNP also localised to the binding site summit of H3K36me3 in primary B cells, LCLs and blood mononuclear cells, with significant, albeit weaker enrichment.

As verification, we performed the same analysis using ChIP-Seq experiments ($n = 612$) from the venous blood portion of the Blueprint Epigenetics consortium[23] (8 modifications across 24 unique cell types from 83 donors). Only 22 ChIP-Seq experiments presented evidence of overlap with the SLE-associated haplotype, and strikingly, all of these intersections were again for the H3K36me3 chromatin modification. No other histone modifications intersected this region. All five SNPs on the 1 kb SLE-associated haplotype were found to overlap with H3K36me3 in monocytes, B cells and neutrophils – corroborating the Roadmap Epigenomics data. We were unable to make robust conclusions on differential H3K36me3 signal between the sexes as the sample sizes per cell-type were too small (Supplementary Fig. 4, Supplementary Table 1).

Lastly, the associated SNPs in the 3′-UTR of *CXorf21* showed no evidence of disrupting a microRNA binding site after interrogation using miRDB[24].

**The risk haplotype interacts with the promoter of *CXorf21***. We sought to investigate a conceivable molecular mechanism whereby the SLE-associated haplotype at the 3′ end of *CXorf21* modulates expression through alteration of chromosome interactions. The promoter capture Hi-C dataset curated by the CHiCP resource[25] was interrogated. This resource comprises Hi-C data from 17 primary immune cell types taken from healthy donors. Three of the five SNPs (rs887369, rs2710402 and rs2532873) on the associated haplotype, which are closest to *CXorf21*, reside within the chrX:30576528–30582605 target region. Across all primary immune cell types tested, the target region was found to interact with four baits (Fig. 3c): the promoter region of *CXorf21* (chrX:30595248–30603761); the promoter of *GK*; and two intronic antisense RNAs of *TAB3* (*TAB3-AS1* and *TAB3-AS2*). Significant bait-target region interactions (CHiCAGO score ≥ 5) were detected exclusively in neutrophils (Fig. 3d), where the *CXorf21* promoter bait interaction presented the greatest strength of interaction with the risk haplotype target region (6.09). Strong but sub-threshold interactions ($3 \leq$ CHiCAGO score $< 5$) were also detected for the risk haplotype target and the *CXorf21* promoter bait region in monocytes (3.72) and naïve B cells (3.15). The strength of the interaction score between the risk haplotype target region and the *CXorf21* promoter was found to correlate strongly with the signal strength of epigenetic marks (from ENCODE[26]) indicative of active gene-expression (H3K4me3 and H3K27ac) for matched cell types (Fig. 3e). These findings suggest that the 3′-promoter interaction of *CXorf21* is more pronounced in the cell types in which *CXorf21* is expressed, and the interaction itself is involved in regulation of expression. In fact, by assessing the transcription factor landscape at the *CXorf21* locus, we found significant binding events of RNA polymerase II (POLR2A) at the 3′ SLE-associated region in immune cell types only; corroborating our hypothesis that the observed chromatin looping is necessary for transcriptional regulation (Supplementary Fig. 5).

**Sexual dimorphic expression is magnified upon activation**. GTEx RNA-Seq data[27] across 45 different cell/tissue types

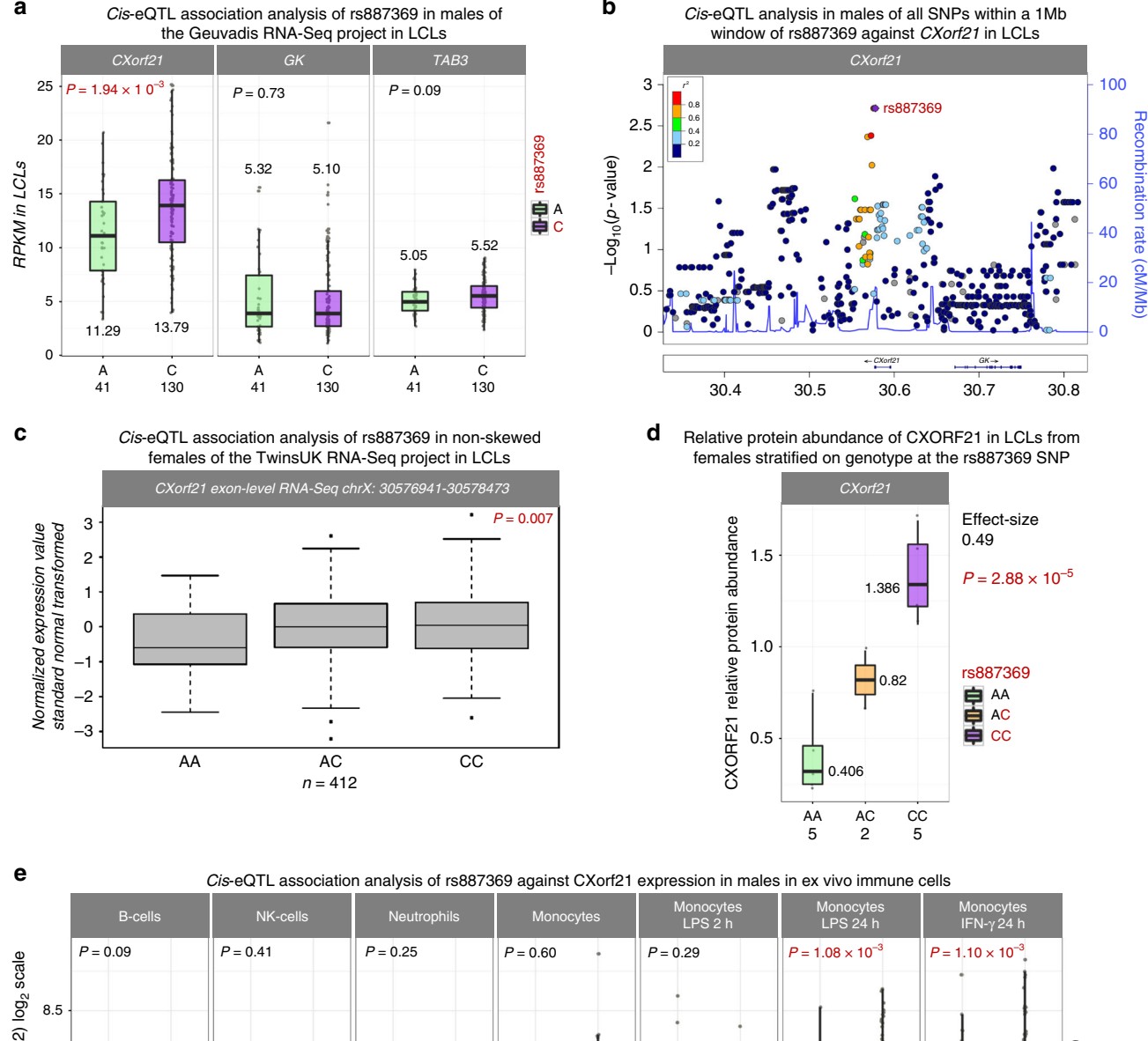

**Fig. 2** eQTL association analysis of SLE associated risk SNP rs887369 in immune cell types. **a** *Cis*-eQTL analysis of rs887369 using male samples of the Geuvadis RNA-Seq expression cohort profiled in LCLs. The MAGEB family of genes and *NR0B1* were not expressed in LCLs (RPKM < 1). Allele [C] of rs887369 tags the risk haplotype. The number underneath each box-plot represents the mean of the group and the number underneath the *x*-axis refers to the number of inviduals in each group. **b** *Cis*-eQTL analysis performed for all SNPs in *cis* (±1 Mb) to rs887369 against *CXorf21* expression using the males of the Geuvadis cohort. The coordinate of each SNP is plotted on the *x*-axis and the −log$_{10}$(P) value of association on the *y*-axis; rs887369 is highlighted as the best eQTL. **c** *cis*-eQTL analysis of rs887369 against *CXorf21* expression in LCLs from the TwinsUK cohort using only females who exhibit non-skewed patterns of X-chromosome inactivation (see methods). **d** Relative protein abundance of CXORF21 in LCLs from females stratified on genotype at the rs887369 SNP. Relative abundance normalised against beta-actin loading control. Source data are provided in the Source Data file (**e**) *Cis*-eQTL analysis of rs887369 using the microarray data from the Fairfax et al.[45,46] and Naranbhai et al.[47] cohorts in primary ex vivo immune cell types (see Methods). The remaining *cis*-genes did not pass quality control. Box-plots show minimum (Q1–1.5*IQR), 25th percentile (Q1), Median, 75th percentile (Q3) and maximum Q3+1.5*IQR

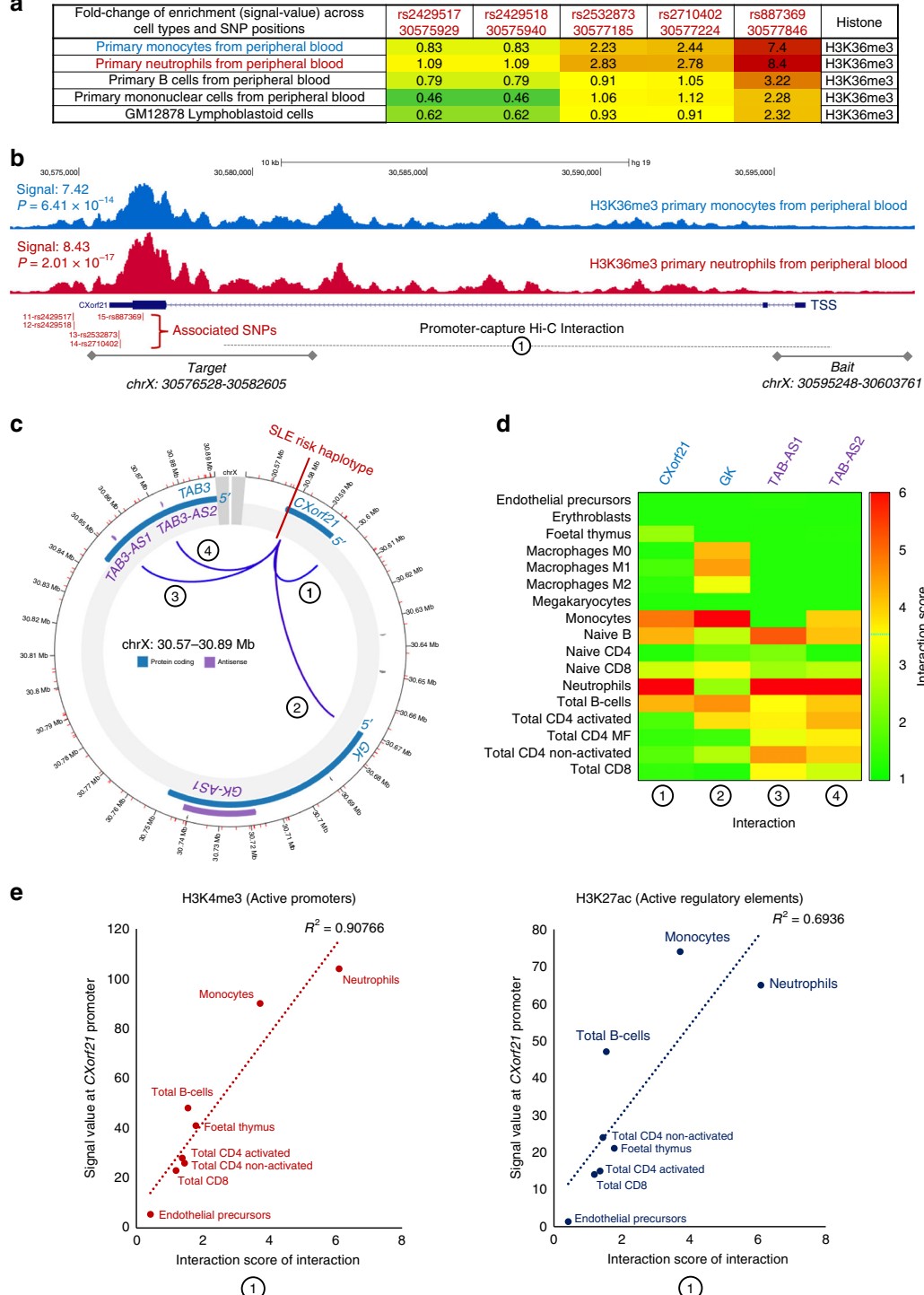

**Fig. 3** Functional prioritisation of causal variants at the Xp21.2 SLE susceptibility locus. The five SNPs carried on the risk haplotype attributed to SLE susceptibility and modulation of *CXorf21* gene expression were epigenetically fine-mapped using chromatin data from the Roadmap Epigenomes Project (twelve different marks across 127 cell/tissue types). **a** The five SNPs localised to significant H3K36me3 modification sites in five immune cell types. The heatmap shows the fold-enrichment of H3K36me3 between cell-types across SNP positions. **b** Signal tracks of H3K36me3 in primary monocytes (blue) and primary neutrophils (red) from peripheral blood across the *CXorf21* susceptibility locus. Only rs887369 localises to the binding site summit of H3K36me3 in these two cell types. **c** Promoter-capture Hi-C interaction of the rs887369 target locus (chrX :30576528–30582605) with four bait loci across 17 primary immune cell types from healthy human donors (Note that the majority of the samples are pooled from multiple donors making it impossible to deconvolute the sex and genotypes of the individuals). Interaction #1 is the interaction between the association target region (at the 3′ end of *CXorf21*) and the *CXorf21* promoter region. **d** Heatmap of strength of interaction (CHiCAGO score) of the four interactions across immune cell types. **e** Correlation of interaction score for interaction #1 (3′ of *CXorf21* and *CXorf21* promoter) with Roadmap Epigenomes Project chromatin marks found at the *CXorf21* promoter across different immune cell types. Higher interactions are correlated with greater enrichment of active chromatin marks suggesting the interaction to regulate gene expression is cell-type specific

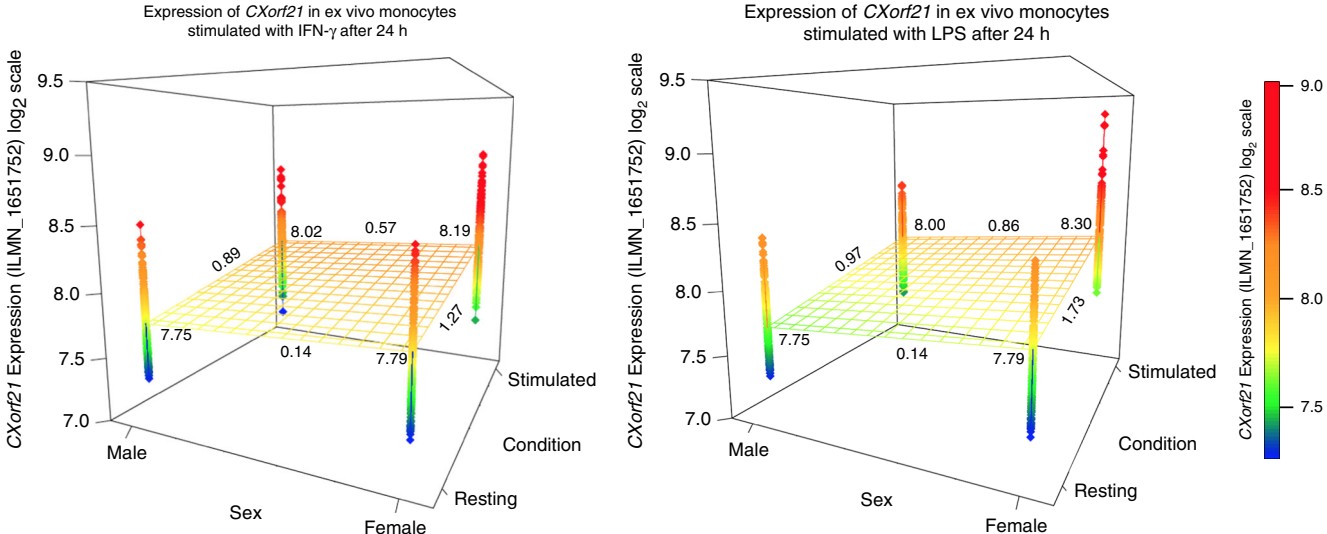

**Fig. 4** Expression of *CXorf21* in primary ex vivo cells stratified on sex and cellular activation. *CXorf21* expression data in resting and stimulated monocytes from healthy individuals of European ancestry derived from the Fairfax et al. studies (see Methods; *n* = 322 biologically independent samples). Samples were separated based on sex and activiation condition: following stimulation with interferon gamma (IFN-γ) or with lipopolysaccharide (LPS) and harvested after 24 h. For each group, the mean is reported in the corners and the effect size (Cohen's d) is reported along the corresponding three-dimensional regression plane. Plots were constructed using plot3D for R. Source data are provided as a Source Data file

confirmed that there is significant sexual dimorphic expression of *CXorf21* in both LCLs and thyroid tissue (LCLs: 1.78-fold greater in females, $P = 1.10 \times 10^{-5}$, thyroid: 1.33-fold greater, $P = 2.65 \times 10^{-3}$ following Bonferroni multiple testing correction; *t*-test; Supplementary Fig. 6a and Supplementary Table 2). Neighbouring genes *GK* and *TAB3* were equally expressed in both sexes, in both LCLs and in the cell types in which both genes are most expressed, suggesting escape from XCI at this locus is restricted to *CXorf21*. Using HapMap LCLs selected on the basis of their genotype at haplotype-tagging rs887369, we employed the validated anti-CXORF21 antibody (Supplementary Fig. 3) to quantify protein abundance by western blot. When we examined cell lines that all carried at least one risk haplotype, we confirmed that protein expression was higher in females (Supplementary Fig. 6b): females harboured 3.6 times more CXORF21 than males ($P = 0.006$; *t*-test). These findings imply that the slight variation in *CXorf21* mRNA results in an amplified effect on overall protein abundance. To ensure these results were not a consequence of monoallelic expression of *CXorf21* in pauciclonal LCLs, we assayed *CXorf21* expression from microarray experiments across a range of primary ex vivo immune cells and found, as with other XCI escaping genes, the effect size of *CXorf21* expression between sexes was cell-type specific[11]. In resting B cells, NK cells, neutrophils and monocytes, no significant difference in transcript abundance of *CXorf21* between sexes was observed (Supplementary Fig. 6c, Supplementary Table 3). However, though we see global increase of *CXorf21* expression in both sexes, a striking sexual dimorphic responses to LPS- or IFN-γ-stimulation in monocytes was observed (Fig. 4; $P_{LPS} = 1.41 \times 10^{-12}$; $P_{IFN-\gamma} = 9.29 \times 10^{-8}$; *t*-test). Transcript abundance of *CXorf21* in monocytes is therefore greatest in females under immune-stimulated conditions.

***CXorf21* is a likely interferon response gene**. Given the marked increase of *CXorf21* expression in stimulated immune cells (including LCLs which exhibit a partially activated phenotype[28]) and the observed up-regulation of IFN-regulated genes in SLE[29], we investigated whether *CXorf21* is an interferon response gene by profiling gene expression using in-house microarray data in primary ex vivo B cells taken from healthy females (*n* = 49 in

total, of which *n* = 32 were treated with IFN-α). We observed *CXorf21* is one of eighteen X chromosome genes (including *TLR7*, *IL13RA1* and *ELF4*) which were up-regulated in response to IFN-α stimulation (fold-change: 2.41; $P = 6.0 \times 10^{-9}$; ANOVA; Fig. 5a). No other Xp21.1 gene was modulated by IFN-α. We profiled the epigenetic landscape surrounding the *CXorf21* locus in ENCODE data and detected significant and localised binding events of NF-κB, STAT1, STAT2, STAT3, IRF4 and IRF3 at the immediate promoter region of *CXorf21* in LCLs (Fig. 5b). We also identified a single interferon-stimulated response element (ISRE) +25bp upstream of the *CXorf21* transcription start site (TSS). This sequence motif and the array of interferon regulatory factors was not detected in any of the promoters of other genes within the Xp21.2 locus (Fig. 5b).

**Functional characterisation of the Xp21.2 SLE risk locus**. Eight genes are encoded at the Xp21.2 SLE risk locus (rs887369; $P = 3.34 \times 10^{-7}$; OR = 1.43): four Melanoma Antigen B (MAGEB) family genes (*MAGEB1-4*), *NR0B1* encoding the DAX1 nuclear receptor, *GK* (glycerol kinase), *TAB3* (TGF-beta activated kinase 1 and MAP3K7 binding protein 3) and *CXorf21* (Fig. 1a; Supplementary Table 4). None of these eight genes had reported associations with immune-related phenotypes in human or mouse.

*CXorf21* is the only gene in the locus with a discrete immune-specific mRNA expression profile; being most highly expressed in the spleen, appendix, bone marrow and lymph nodes (Protein Atlas; Supplementary Fig. 7, GTEx and FANTOM5 validation in Supplementary Fig. 8). This suggests the mechanism by which the SLE-risk haplotype is affecting disease risk is through candidate gene *CXorf21*. To refine this analysis in terms of cellular expression, we used data from Blueprint Epigenome (RNA-sequencing) and BioGPS (microarray) to show that within immune cell types, the expression of *CXorf21* is largely restricted to monocytes, neutrophils and B cells (Supplementary Fig. 9). We corroborated these findings in RoadMap Epigenomics data and found a striking chromatin landscape around the transcription start site of *CXorf21*, indicative of epigenetic silencing in non-immune cell types (Supplementary Fig. 10). The expression

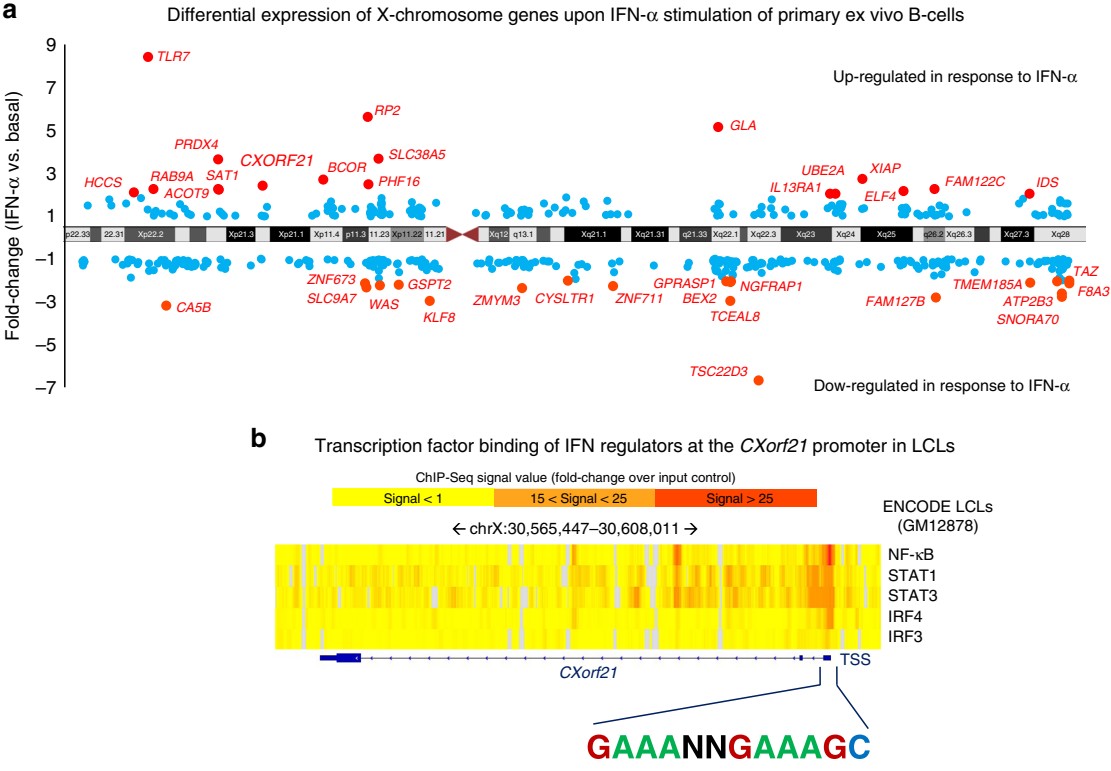

**Fig. 5** CXorf21 as an interferon response gene. **a** Differential gene expression of X chromosome genes in response to IFN-α stimulation (harvested after 6 h) in primary ex vivo B cells from healthy females of European ancestry (in-house data). Genes highlighted in red are significantly differentially expressed (q < 0.01; absolute fold-change >2). **b** Epigenetic landscape of *CXorf21* using ENCODE transcription factor binding data in LCLs (GM12878 cell line). All five transcription factors have genome-wide significant binding sites at the *CXorf21* promoter. Heat colour is a function of signal strength (fold-change over input)

profile of *CXorf21* at protein-level was largely consistent with the mRNA data; though CXORF21 protein was found to be in equal abundance in certain secondary immune tissue such as the bowel and skin (Supplementary Fig. 11).

RNA-Seq co-expression analysis across a range of human cell and tissue-types was undertaken using the COEXPRES algorithm[30]. The results indicate that *GPR65* (G-couple protein receptor 65) tops the ranking, whose protein product is important in lysosomal function[31]. Examination of the top 100 ranked genes revealed the expression signature of *CXorf21* correlated with the Toll-like receptor (TLR) signalling pathway including *TLR7*, *TLR6*, *PIK3CG* and *PIK3CD* (Supplementary Table 5)[30], of which *TLR7* was highest ranked. The correlation between the expression of the two X-linked genes, *CXorf21* and *TLR7*, was replicated in TwinsUK RNA-Seq data[21] from LCLs from non-skewed females (n = 271; $\rho$ = 0.38; P = 6 × 10$^{-11}$).

In order to gain further insight into the potential function of CXORF21, we utilised high-throughput affinity-purification mass spectrometry data from BioPlex[32] and revealed a high confidence (quantitative score: 0.999) protein–protein interaction between CXORF21 and SLC15A4, encoded by the SLE susceptibility gene *SLC15A4* (rs1059312; $P_{META}$ = 1.48 × 10$^{-13}$; OR = 1.17)[15]. *SLC15A4* is an immune-restricted lysosomal amino-acid transporter required for TLR7- and TLR9-mediated type I IFN production in dendritic cells and B cells in lupus[33]. Interestingly, in the BioPlex data, CXORF21 was also found to interact with itself, suggesting probable oligomerization of this protein.

**Protein level correlates with disease activity in females**. In a modest cohort ($n_{cases}$ = 19; $n_{controls}$ = 13) we did not observe a statistically significant difference in CXORF21 protein abundance

between female case and controls in CD14+ monocytes or CD19+ B cells (Supplementary Fig. 12). However, we observed an age-dependent correlation between CXORF21 and SLE Disease Activity Index (SLEDAI). CXORF21 protein abundance is positively correlated with SLEDAI in SLE females <35 years of age (CXORF21 ~ SLEDAI * Stratified Age) in both CD14+ monocytes and CD19+ B cells (Supplementary Fig. 13). A likelihood ratio test (LRT) rejected the model of SLEDAI as a single variable (upper panels Supplementary Fig. 13) in favour of an interaction model in monocytes (LRT P = 0.0002) and B cells (LRT P = 0.0006). The rejection of the single variable models are also supported by BIC ($\Delta BIC_{monocytes}$ = 8.1; $\Delta BIC_{Bcells}$ = 5.9). We observed a significant interaction term (SLEDAI * Stratified Age) in monocytes (P = 0.002), though the interaction term in B cells did not pass multiple testing correction (P = 0.011; lower panels Supplementary Fig. 13).

**CXORF21 protein may act within endosomal pathway**. CXORF21 is a small protein of ~34 kDa as identified by Western Blot. Very little of the secondary/tertiary protein structure of CXORF21 could be accurately determined by the Phyre bioinformatics prediction tool[34]. Thus, to gain insight into the protein's function we sought to determine its cellular location in ex vivo cells from healthy females and the GM12878 lymphoblastoid cell line. We undertook multispectral imaging flow cytometry (MIFC) with a range of labels for different organelles. The results demonstrated minimal co-localisation of CXORF21 with nuclear, Golgi or lysosomal markers in ex vivo PBMCs, and this was not affected by IFN stimulation (Supplementary Figs. 14 and 15). In view of these negative findings and the data showing co-expression of *CXorf21* with components of the Toll-like

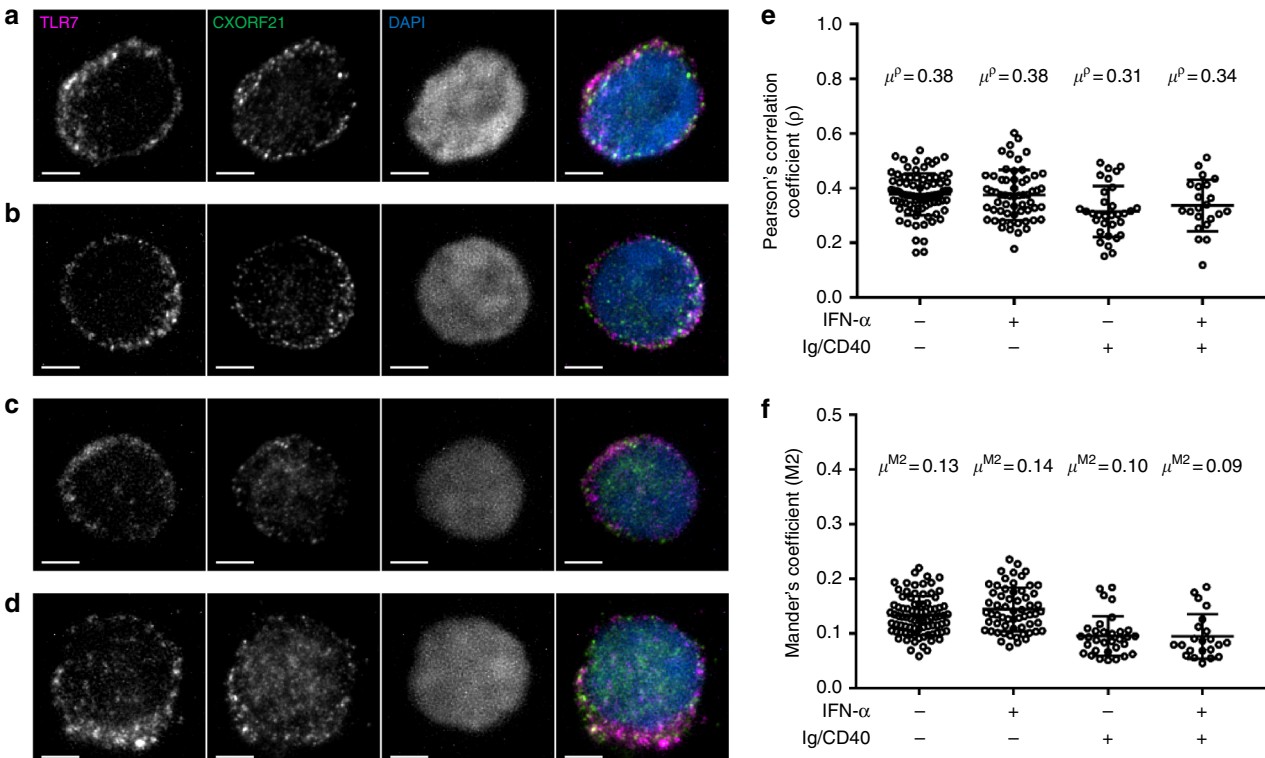

**Fig. 6** Super resolution microscopy of CXORF21 and TLR7. Structured Illumination Microscopy data showing colocalisation of TLR7 and CXORF21 in ex vivo B cells. Representative results on individual cells are shown in **a** through D with TLR7 staining in the first column, CXORF21 in the second column, DAPI nuclear staining in column three, and in the fourth column all three stains are merged: TLR7 (magenta), CXORF21 (green) and DAPI (blue). The B cells are under different conditions in the panels: **a** resting, **b** resting and IFN-α treated (1000 U/ml), **c** Ig/CD40 stimulated and **d** Ig/CD40 stimulated and IFN-α treated ex vivo B cells at 20 h. Maximum intensity projections are shown. Scale bar in white on bottom left hand corner is 2 μm. **e** Plot showing the correlation co-efficients ($\rho$) between TLR7 and CXORF21 staining of multiple B cells quantified using the results from Z-stack images from individual cells (represented as open circles). From left to right: unstimulated cells ($n = 84$), cells stimulated with IFN-α ($n = 60$), B cells stimulated with Ig/CD40 ($n = 32$), B cells stimulated with Ig/CD40 and IFN-α ($n = 22$). The horizontal bar represents the mean correlation co-efficient ($\mu^\rho$) and the bars above and below this denote the standard deviation of the distribution. **f** Mander's colocalisation coefficient (M2) between TLR7 and CXORF21 are shown from Z-stack images from single B cells (represented as open circles). From left to right: unstimulated cells ($n = 84$), cells stimulated with IFN-α ($n = 60$), B cells stimulated with Ig/CD40 ($n = 32$), B cells stimulated with Ig/CD40 and IFN-α ($n = 22$). The horizontal bar represents the mean colocalisation co-efficient ($\mu^{M2}$) and the bars above and below this denote the standard deviation of the distribution. Source data are provided as a Source Data file

receptor signalling pathway (Supplementary Table 6), we utilised the increased resolution of structured illumination microscopy (SIM) to determine whether there was any evidence for colocalisation of CXORF21 with TLR7. Representative images for the staining in resting and stimulated ex vivo B cells are shown (Fig. 6a through 6d). We quantified the correlation between signals obtained from CXORF21 with TLR7 staining (Fig. 6e) and determined the colocalisation of the two staining signals in B cells using Mander's co-efficient (see Methods). These analyses were undertaken in both resting B cells and stimulated B cells (B cell receptor cross-linking and CD40) and in each case with and without exposure to IFN-α. We conclude that there is weak colocalisation between TLR7 and CXORF21 in ex vivo B cells (Pearson correlation $0.3 < \rho < 0.4$). No significant differences in colocalisation between CXORF21 and TLR7 were observed after IFN-α treatment of resting or IgM/CD40 stimulated B cells.

As the endosomal intracellular pathway interacts with the autophagy pathway (which has also been implicated in SLE pathogenesis)[35] we sought to determine whether CXORF21 colocalised with autophagosomes, once more utilising SIM. Using LC3 as a marker of the autophagosome, representative results of the joint staining (LC3 and CXORF21) are shown for Ig/CD40 stimulated B cells (Fig. 7a) with exposure to the inhibitor of autophagic flux, bafilomycin (Fig. 7b) and Ig/TLR7/8 stimulated

B cells (Fig. 7c) with bafilomycin (Fig. 7d). The results from multiple cells are summarised in Fig. 7e, f, which show no colocalisation between LC3 and CXORF21 in bafilomycin-treated ex vivo B cells when stimulated with Ig/CD40 or Ig/TLR7/8. Assaying CXORF21 protein abundance by western blot in starved LCL (see methods) indicates that the amount of CXORF21 is not altered by the addition of bafilomycin and hence it is unlikely that CXORF21 is an autophagy substrate (Fig. 7g; left panel). The blot shows some elevation of sequestome 1 (p62), an autophagosome cargo protein, following exposure to bafilomycin, which would be expected (Fig. 7g; right panel).

## Discussion

The underrepresentation of genetic associations on the X chromosome in autoimmune disease is highly paradoxical given the prominent sex bias towards females and the increased density of immune related genes compared to the autosomes. This is partly due to the paucity of sex chromosome data in genome-wide studies; only 33% of GWAS report sex chromosome data[36]. We sought to functionally investigate the undefined SLE susceptibility locus Xp21.2 from our own GWAS dataset (rs887369; $P = 3.34 \times 10^{-7}$; OR = 1.43). Our investigation defines *CXorf21* – encoding a protein of hitherto unknown function – as the candidate gene and

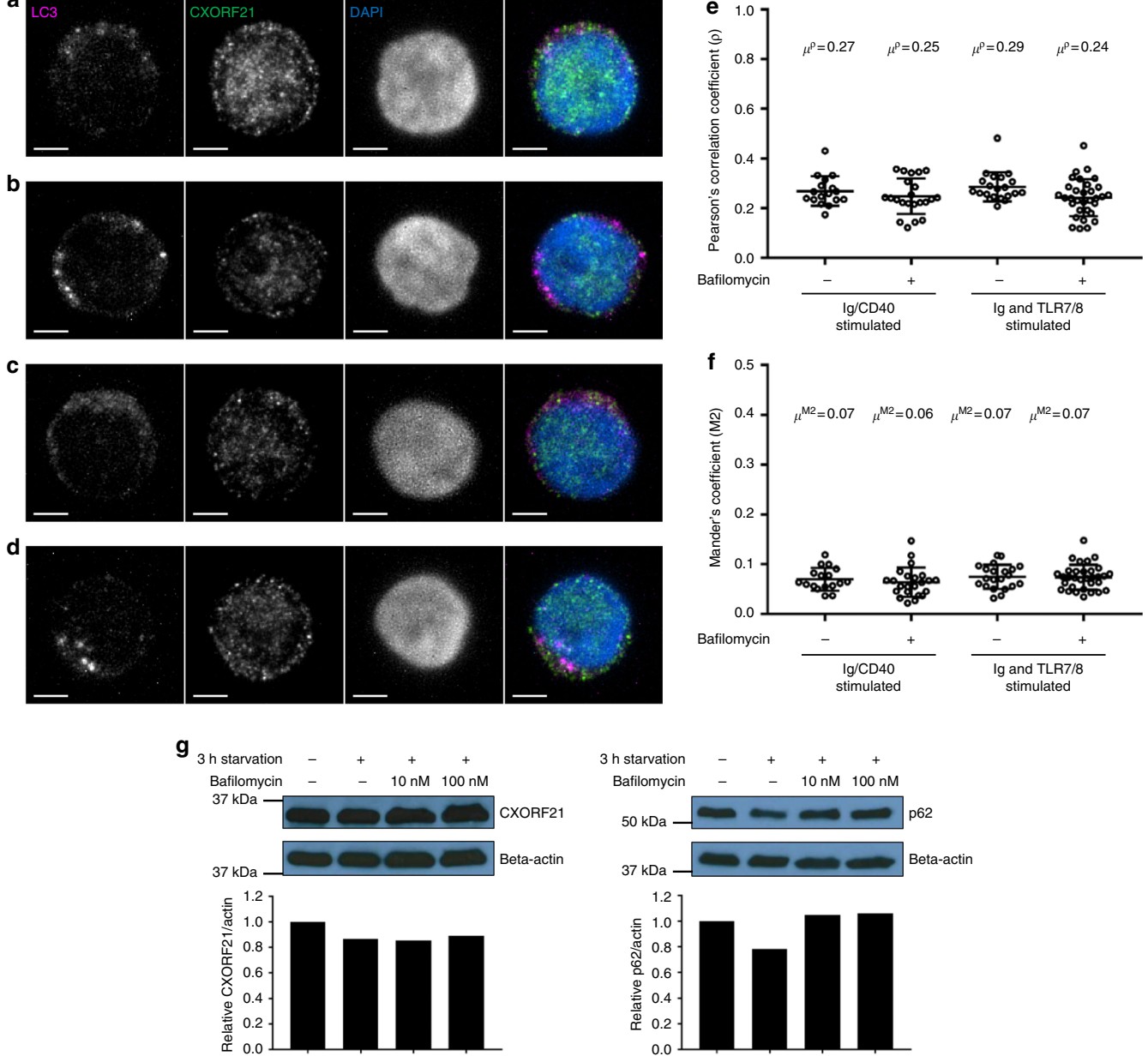

**Fig. 7** CXORF21 and the autophagosome. Structured Illumination Microscopy data showing colocalisation of LC3 and CXORF21 in ex vivo B cells. Representative results on individual cells are shown in panels **a** through **d** with LC3 staining in the first column, CXORF21 in the second column, DAPI nuclear staining in the third column, and in the fourth column all three stains are merged: LC3 (magenta), CXORF21 (green) and DAPI (blue). In panel **a** B cells were Ig/CD40 stimulated, **b** Ig/CD40 stimulated and bafilomycin-treated, **c** Ig and TLR7/8 stimulated and **d** Ig and TLR7/8 stimulated and bafilomycin-treated after 27 h. Maximum intensity projections are shown. Scale bar in white on bottom left hand corner is 2 μm. **e** Plot showing the correlation co-efficients ($\rho$) between LC3 and CXORF21 staining quantified using the results from Z-stack images, individual B cells are represented as open circles. From left to right: Ig/CD40 stimulated cells ($n = 17$), Ig/CD40 stimulated cells in the presence of 10 nM bafilomycin ($n = 22$), B cells stimulated with Ig and resiquimod ($n = 21$), B cells stimulated with Ig and resiquimod in the presence of 10 nM bafilomycin ($n = 32$). The horizontal bar represents the mean correlation co-efficient ($\mu^{\rho}$) and the bars above and below this horizontal bar denote the standard deviation of the distribution. **f** Mander's colocalisation coefficient (M2) between LC3 and CXORF21 are shown from Z-stack images, individual B cells are represented as open circles. From left to right: Ig/CD40 stimulated cells ($n = 17$), Ig/CD40 stimulated cells in the presence of 10 nM bafilomycin ($n = 23$), B cells stimulated with Ig and resiquimod ($n = 21$), B cells stimulated with Ig and resiquimod in the presence of 10 nM bafilomycin ($n = 32$). The horizontal bar represents the mean colocalisation co-efficient ($\mu^{M2}$) and the bars above and below this denote the standard deviation of the distribution. **g** Western blot analysis of protein extract from starved LCL, in the left-hand blot CXORF21 is quantified in the absence of bafilomycin and after 10 nM and 100 nM treatment. The amount of CXORF21 was quantified by densitometry and the relative abundance shown against a beta actin control, using the unstimulated conditions as a reference point. In the right-hand blot sequestosome 1 (p62) is quantified in the absence of bafilomycin and after 10 nM and 100 nM treatment. The amount of Sequestosome-1 was quantified by densitometry and the relative abundance shown against a beta actin control, using the unstimulated conditions as a reference point. Source data are provided as a Source Data file

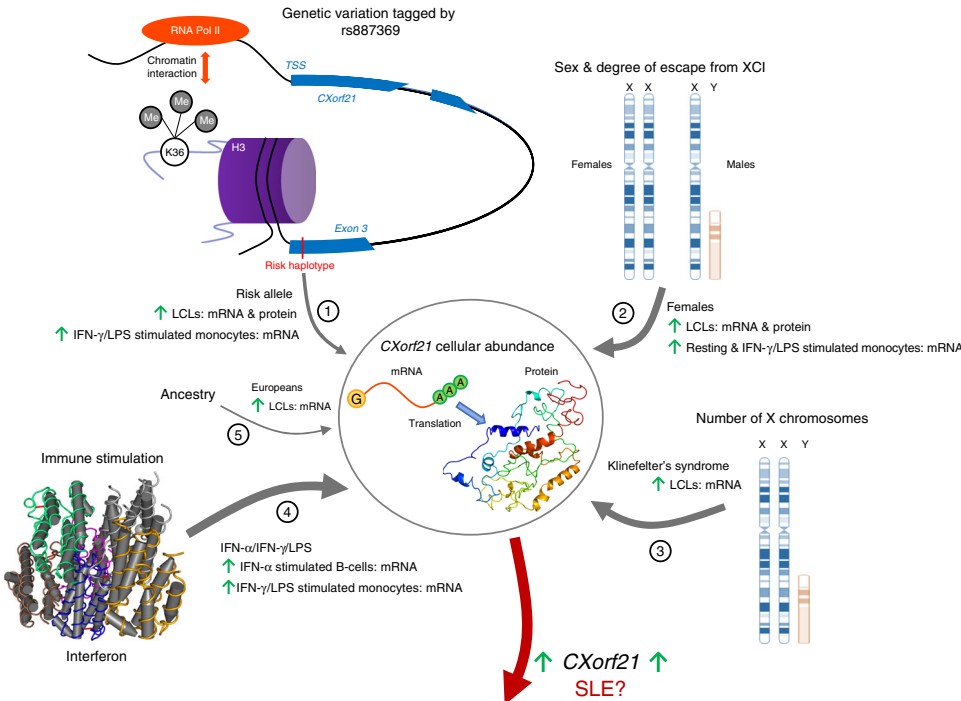

**Fig. 8** Summary of factors influencing expression of *CXorf21* at RNA and protein level. We summarise five factors increasing the cellular abundance of *CXorf21* either at RNA level or protein level across a range of immune cell types. These are: (1) genetic variation at SLE susceptibility haplotype - tagged by SNP rs887369 - where the risk haplotype [C] may drive up-regulation by modulation of chromatin interaction and/or modification of H3K36me3 state; (2) female sex, in which escape from X-inactivation results in an increased amount of transcript and protein in females; (3) X chromosome aneuploidy; (4) type I and type II interferons, and LPS, increase the expression of *CXorf21* in ex vivo B cells and monocytes; (5) ancestry – potentially linked to the minor allele frequency of rs887369 – in which higher levels of *CXorf21* transcript is observed in LCLs derived from donors with European ancestry. We hypothesise that elevation of CXORF21 is a risk factor for developing SLE and that this is may be mediated through it's role in the endosomal pathway. Figure generated by C.A.O

demonstrates its expression is upregulated through by a number of distinct factors: chromosome X dosage and loss of XCI, the risk haplotype (tagged by rs887369) and cellular activation by interferon (summarised in Fig. 8). Our study supports the hypothesis that altered expression of X-linked genes contributes to the sexual dimorphism in autoimmunity[14] and provides some preliminary evidence for the role of *CXorf21* in SLE, although this topic clearly warrants further investigation.

To date, six X-encoded SLE susceptibility loci have been identified, and four have been shown to harbour genes that escape XCI (*TLR7*, *TMEM187*, *IRAK1*, and *CXorf21*). Of these, *CXorf21* is the most robustly escaping; possessing evidence of escape in ~80% of individuals in contrast to the remaining genes that exhibit escape in <30%[16]. We show that escape from XCI is highly localised to *CXorf21* across a ±1-Mb window.

The Xp21.2 locus is not as strongly associated with SLE in individuals of non-European ancestry, although an association has been reported in Koreans[37]. This is partially explained by the marked disparity in allele frequency of risk allele rs887369 [major allele: C] between populations (1000Genomes: 0.76 in Europeans, 0.92 in Africans, 0.95 in Asians). The lower minor allele frequency in non-Europeans may clearly impact on power, especially as non-European GWAS have been of smaller sample size. The fraction of individuals who exhibit XCI of *CXorf21* is reported to be diminished in individuals of African descent (relative to those of European descent[16]); however, lower allele frequencies of transcribed polymorphisms and limited samples sizes impede power. Whether allele frequency of rs887369 and reduced XCI escape are correlated or whether variation at rs887369 itself is causal to a degree of escape poses an interesting line of enquiry. Furthermore, the reduced level of escape in non-

Europeans may mean the effect size will limit the power of this locus to be detected.

*CXorf21* has a discrete expression pattern in immune cells, both adaptive and innate, with the greatest expression of *CXorf21* found in monocytes and neutrophils, primary B cells and LCLs. It appears to be epigenetically inert in non-immune cell types, suggesting the regulatory mechanisms driving expression of *CXorf21* are not present in non-immune cell types. These data align with the observation that other candidate genes of SLE and their accompanying causal variants exhibit a discrete expression signature and *cis*-regulatory landscape that is largely restricted to immune cell subsets; particularly B cells (including B-lymphoblastoid cell lines), T cells and monocytes[15,38–40].

We demonstrate that *CXorf21* expression is upregulated in LPS and IFN-γ-stimulated monocytes, and in IFN-α-stimulated B cells, with the magnitude of increase greater in females leading to significant sexually dimorphic expression levels. We have also identified binding sites of respective transcription factors from these signalling cascades: IRF3, NF-κB and STAT1-3 at the immediate promoter of *CXorf21* suggesting *CXorf21* transcription could be a primary response gene of the TLR4 (IRF3) and IFN (STATs) signalling pathways. However, *CXorf21* expression decreases following acute (2 h) LPS-stimulation, suggesting *CXorf21* is in fact a late response gene induced by secondary activation of the TLR4-induced type I IFN feed forward loop[41]. Indeed, late response genes are characterised by STAT binding sites and ISRE[41], which we also identify in the *CXorf21* promoter. Sex differences in the LPS-induced monocyte response have been previously reported, whereby females have heightened activation and cytokine release compared with males, although the underlying mechanism has yet to be delineated[42,43].

rs887369 tags a short 1-kb haplotype comprising five perfectly correlated SNPs. The haplotype is an eQTL for *CXorf21*, with the risk allele increasing the gene's expression; we hypothesise that a self-regulatory mechanism involving modification of H3K36me3 state and chromatin looping affects RNA polymerase II within the gene promoter (Fig. 8).

The expression of *CXorf21* transcript has previously been shown to be the most accurate delineator of disease flare from infection in SLE patients[18]. Interestingly, this previous study was conducted in largely non-European patients, suggesting the role of *CXorf21* is not limited to individuals of European ancestry. Further supporting our hypothesis that *CXorf21* is an IFN-inducible gene, the genes with dysregulated expression at exome-wide significant expression changes identified in this study are enriched for IFN-inducible genes[18]. We observed an age-dependent correlation between CXORF21 expression and disease activity using flow cytometry in a modest cohort, with CXORF21 protein abundance positively correlating with SLEDAI in patients <35 years of age. These data warrant further investigation and suggest age-stratified analysis in disease cohorts could be illuminating.

The CXORF21 protein has no known function and the primary amino acid sequence gives no clear clues in this regard. In an attempt to provide some insight into the function of CXORF21, we conducted a number of imaging studies to investigate its intracellular location. These studies showed that CXORF21 is present in both the nucleus and cytoplasm. Interestingly, we show that there was some colocalisation of CXORF21 with TLR7 in B cells. This intracellular toll-like receptor was selected for imaging as it is known to play a role in nucleic acid sensing in SLE and our analyses revealed some degree of co-expression of *TLR7* and *CXorf21* at the RNA level. Intracellular toll-like receptors operate in a complex system involving the endosomal and lysosomal compartments[44]. However, the precise points at which CXORF21 and TLR7 may interact within these compartments is not clear on the basis of our data, but further exploration of this question should reveal more insights into the function of CXORF21 and how it promotes systemic autoimmunity.

The female-biased sex imbalance of autoimmune diseases is not understood. Our study, which characterises the SLE association at *CXorf21*, an IFN-inducible gene which escapes XCI, adds support to the hypothesis that sex bias in immune function has a genetic basis and provides an underlying immunological mechanism that underpins the sexual dimorphism in SLE.

## Methods

**European SLE GWAS data.** Genotype data from 10,995 individuals of matched European ancestry (4036 SLE cases, 6959 controls) genotyped on the Illumina HumanOmni1 BeadChip from the Bentham and Morris et al.[15] study were imputed as outlined below. These data had undergone quality control and PCA as described[15].

**Imputation.** The European SLE GWAS[15], Fairfax et al.[45,46] and Naranbhai et al.[47] cohorts were imputed using UK10K-1000GP3 merged reference panel across the X:30077468–31077846 1-Mb region, plus a 2-Mb buffer region (GRCh37 assembly). A full imputation without pre-phasing was conducted using IMPUTE2 to increase the accuracy of imputed genotype calls[48,49]. Imputed genotypes were filtered using an info score threshold of 0.5. The most likely genotype from IMPUTE2 was taken if its probability was >0.5. If the probability fell below this threshold, it was set as missing.

**Allelic and haplotype association fine-mapping.** Imputed data from the European SLE GWAS were filtered to include variants with MAF > 0.01 and HWE > $1 \times 10^{-4}$, and minimum genotype rate >90%. SNPTEST 2.5.2[50] was used to test for additive models of allelic associations across the X:30077468–31077846 1 Mb region, fitting a logistic regression model (including the first four covariates from the original GWAS[15]) with equal effect size between males and females[50,51]. Independent signals were assessed by including the genotype for the rs887369 SNP as a covariate using the SNPTEST algorithm. Association plots were generated

using LocusZoom[52]. Haplotype association analysis and LD calculations between SNPs were performed using Haploview 4.2[53] (implementing X-chromosome analysis) using the entire GWAS of 10,995 individuals. Specifically, haplotype blocks across a 100-kb region anchored on rs887369 were defined by the confidence internals algorithm[54] and haplotype association testing performed by a $\chi^2$ test using marker thresholds of MAF > 0.01 and HWE > $1 \times 10^{-4}$, and minimum genotype rate >90% (657 SNPs in total).

We fitted two models for association in SNPTEST. The inactivation model is the default in SNPTEST's newmlmethod with male genotypes coded as 0/1 and females coded as 0/0.5/1 and one shared estimated effect (log odds ratio). In the escape model we used SNPTEST with the stratify_on option which fits separate effects for males and females. In both models, we fit a different intercept for males and females (using sex as a covariate in the inactivation model) and so the two models only differ by one parameter (being the differing log odds ratio). A likelihood ratio test (LRT) on one degree of freedom was performed in R (using the likelihood values output by SNPTEST), where the escape model was tested against the simpler inactivation model. A statistically significant result (based on the p-values form the LRT) would therefore reject the inactivation model.

**Genotype data for ex vivo cell eQTL cohorts.** X chromosome SNPs of the Fairfax et al.[45,46] and Naranbhai et al.[47] cohorts with an Illumina GenCall score of <0.7 and called on both X and Y were removed. PLINK v1.9[55] was used to remove samples that failed sex check assignments. Following separation of male and females, SNPs were removed if: HWE < $1 \times 10^{-4}$, MAF < 0.01 and SNP missingness >10%. Individuals were removed if >10% of SNPs were missing. Coordinates were converted from hg18 to hg19 using the UCSC liftOver application[56].

**Genotype-expression cohorts and *cis*-eQTL analysis.** Gene-level RNA-Seq data from LCLs were downloaded from ArrayExpress (Geuvadis; EGEUV-1)[57] and genotypes (X:30077468–31077846) of these individuals containing SNPs (MAF > 0.05) were taken from the 1000 Genomes Project Phase III[58]. Expression data of purified ex vivo primary immune cells were obtained from Fairfax et al.[45,46] and Naranbhai et al.[47] Details are described in the respective articles. Data include resting B-cells, natural killer cells and monocytes;[45] IFN-γ stimulated monocytes after 24 h, LPS stimulated monocytes after 2 h, LPS stimulated monocytes after 24 h;[46] and resting neutrophils[47]. In all instances, *cis*-eQTL association analysis (1 Mb of rs887369) was performed against expression residuals using the linear-model of the MatrixeQTL R package[59] including the number of PCs described in the respective articles.

The TwinsUK RNA-Seq eQTL cohort profiled in LCLs[21] was used for *cis*-eQTL association analysis in non-skewed females (n = 412). Individuals were firstly assessed for skewed X-chromosome inactivation patterns using allele-specific expression of *Xist* to estimate the proportion of X inactivation from each parental X chromosome. Individuals were removed if the allele-specific expression of XIST-linked SNPs was <0.2 or >0.8, these parameters were chosen on the basis of precedence[60–62]. *cis*-eQTL analysis in the twins was performed as above against exon-count residuals corrected for probabilistic estimation of expression residuals (PEER) factors and family relatedness[63].

Differential expression analysis of *CXorf21* between males and females using GTEx RNA-Seq (TMP) data across the 45 cell/tissue types where expression data were available for both sexes was performed using an unpaired *t*-test between males and females after grouping by cell/tissue type. Associations passing the Bonferroni adjusted *P*-value cut-off of $P_{BF} < 0.05$ were deemed significant.

**Cell culture.** LCLs were obtained from Coriell Biorepository and cultured in suspension at 5% $CO_2$, 37 °C in RPMI 1640 medium supplemented with 2 mM L-glutamine, 15% foetal bovine serum, 100 units/ml penicillin and 100 μg/ml streptomycin. Cells were seeded every 2 days to a concentration of 300,000 viable cells/ml. Individuals used in functional assays were of European descent (GM12878, HG01702, HG01786, HG01746, HG0111, HG01628, HG00254, HG12878, HG01501, HG01507, HG01504, HG00269 and HG00232).

**qPCR.** Total RNA was extracted with the RNeasy Mini Kit (QIAGEN) according to manufacturer's instructions. cDNA synthesised with the cDNA Synthesis Kit (Thermo Scientific) and quantified using the NanoDrop 2000 spectrophotometer. qPCR reactions performed using the TaqMan® Universal PCR Master Mix and Universal ProbeLibrary System Technology (UPL) from Roche. Primers were purchased from Sigma and reactions performed using the Applied Biosystems 7500 and subsequent analysis with SDS 2.3. *CXorf21* F: GGATGTTTGACACA-GACTTCAAA, R: CCGGATCAGATGAGCAGATT, UPL #65. *ACTB* F: AGAGCTACGAGCTGCCTGAC, R: CGTGGATGCCACAGGACT, UPL #9. Relative abundance and fold change was calculated using the ΔΔCt method.

**Verification of anti-CXORF21 antibody by gene-knockdown.** Gene-knockdown of *CXorf21* in LCLs (GM12878) was performed by siRNA using the Nucleofector II Device (Lonza) and Amaxa Cell Line Nucleofector Kit V. Two days before transfection, cells were seeded to a concentration of $0.5 \times 10^6$ cells/ml. In duplicate, $2 \times 10^6$ cells were spun at 100 g for 10 min and re-suspended in 100 μl supplemented transfection solution and 20 pmol Silencer Select Pre-Designed &

Validated siRNA (Thermo Fisher Scientific) against *CXorf21* (#4392420). The Silencer Select Negative Control No. 1 siRNA (#4390843) was used as a non-targeting negative control at the same concentration. Cell/siRNA suspensions were transferred to a Nucleofector cuvette and electroporated using the X-001 programme. Samples were cultured in 1.5 ml medium in a 12-well plate and harvested 48 h post-transfection.

**Immunoblot**. Cell lysates were prepared in RIPA buffer (Sigma-Aldrich) and run on a SDS polyacrylamide gel for electrophoresis. Protein was transferred onto a nitrocellulose membrane and blocked in 5% milk-PBS solution. The rabbit polyclonal anti-CXORF21 antibody (Atlas Antibodies; HPA001185) was used at a concentration of 1:1000 and the secondary polyclonal swine anti-rabbit immunoglobulins/HRP (Dako; P0217) at 1:1000. Membranes were stripped by Restore™ Western Blot Stripping Buffer (Thermo Fisher) and re-probed with mouse monoclonal β-Actin antibody (Santa Cruz Biotechnology; sc-47778) at 1:4,000 and anti-mouse IgG HRP conjugate (Promega; W4028) at 1:5,000 or secondary goat anti-mouse IgG HRP conjugate (Invitrogen; A16078) at 1:10,000. ImageJ was used to calculate the density of the bands relative to the loading control. Rabbit anti-SQSTM1/p62 (Cell Signalling, 5114) was used at a concentration of 1:1000 and detected with secondary goat anti-rabbit IgG HRP conjugate (Invitrogen; A16110) at 1:10,000. Raw blots are presented in accompnying Source Data file.

**Epigenetic fine-mapping**. SNPs in X:30077468–31077846 were downloaded from the 1000 Genomes Project Phase III[58]. Epigenetic data across all available cell types ($n = 127$) in NarrowPeak format were obtained from the NIH Roadmap Epigenomics Project[22]. Peaks were filtered for genome-wide significance using an FDR threshold of 0.01, and peak widths harmonised to 2 kb in length centred on the peak summit. SNPs were reported as being localised to an epigenetic mark if they overlapped the 2 kb region. The signal value of the epigenetic mark was reported for the exact coordinate of the SNP using the signal track of the mark in bigWig format visualised using IGV v2.3.80[64].

NarrowPeak files of ChIP-Seq experiments (H3K4me3, H3K27ac, H3K4me1, H3K36me3, H3K27me3, H3K9me3, H3K9/14ac and H2A.Zac) were downloaded from the Blueprint Epigenome Project ftp site (http://ftp.ebi.ac.uk/pub/databases/blueprint/data/homo_sapiens/GRCh38/). Only non-diseased cell-types from venous blood were selected for analysis (24 unique cell-types). Using the GRCh38 genomic positions of the 5 SNPs carried on the associated haplotype, intersection was performed against the genome-wide binding sites of the selected Blueprint ChIP-Seq experiments as per the Roadmap Epigenomics project (above). Fold-enrichment of the peaks that overlapped the associated haplotype were compared by unpaired *t*-test between males and female samples for H3K36me3 across different cell-types.

**Promoter capture Hi-C chromatin interaction data**. Chromatin interaction data across a 17 primary immune cell-types was assessed using Capture Hi-C Plotter (CHiCP; www.chicp.org)[25]. The study focuses on autoimmune susceptibility loci from GWAS and ImmunoChip integrating promoter capture Hi-C datasets from three separate studies[65–67]. The bait to target coordinates and interaction scores were extracted from CHiCP manually. Scores were defined by the CHiCAGO algorithm[68], where scores ≥5 were considered as significant interactions.

**B-cell isolation and cell stimulation**. CD19+ B cells from healthy female subjects ($n = 49$) were isolated by negative selection using the Dynabeads Untouched Human B Cells Kit (Invitrogen). 1.5–3 × 10^6 cells/ml ex vivo B cells were cultured in RPMI 1640 medium, supplemented with 20% FCS, 2mM L-glutamine and 100 U/mL penicillin/streptomycin. B cells from 32 of the 49 subjects were incubated with or without IFN-α 2b (1000 U/mL; PBL Assay Science) at 37 °C and 5% CO$_2$. Cells were harvested after 6 h or 20 h as indicated.

For immunostaining, human B Cell Isolation Kit II (Miltenyi Biotec). Ex vivo B cells (1 × 10^6 cells/ml) were cultured in RPMI 1640 medium, supplemented with 10% heat-inactivated FBS, 2 mM L-glutamine and 100 U/mL penicillin/streptomycin. B cells were stimulated with 10 μg/ml F(ab')2 Fragment Anti-Human IgG+IgM (Jackson ImmunoResearch) and either 0.1 μg/ml CD40L with 0.1 μg/ml Enhancer (Enzo) or 5 μg/ml resiquimod (Sigma). B cells were incubated with or without 10 nM bafilomycin A1 (Sigma) for 3 h before harvesting and with or without 1000 U/mL IFN-α 2b (PBL Assay Science) at 37 °C and 5% CO$_2$. Cells were harvested after 20 h or 27 h as indicated.

**Ex vivo B cell RNA extraction and array hybridisation**. RNA was isolated using the RNeasy Mini kit (Qiagen) according to the manufacturer's instructions and integrity assessed using the Agilent 2100 Bioanalyzer (Agilent) with the RNA 6000 Pico Kit (RIN < 8 excluded). cDNA was synthesised from 50ng of RNA using the High Capacity RNA-to-cDNA Kit (Applied Biosystems). Each sample was hybridised to Affymetrix Human Exon 1.0 ST arrays and expression data were obtained by fluorescence-based detection using the GeneChip Scanner 3000 7G (Affymetrix). Signal intensities were quantified and stored as CEL files.

**Quality control of exon array**. Quality control was carried out using the probe-set and transcript cluster annotation release 33.1 (GRCh37 build). Probe and probe-set filters were applied to the data as recommended[69]. All probe sets targeting RefSeq-annotated RNA transcripts were included. Probes containing polymorphisms (MAF > 0.01) from 1000Genomes were removed. Cross-hybridising probes and probe sets containing less than three probes were also excluded. Detection above background noise (DABG) was calculated for all CEL files and probe-sets were filtered using Affymetrix Power Tools. Probe sets with DABG $P > 0.01$ in 50% of resting or IFN-α stimulated samples were removed. Probes and probe-sets that failed QC filters were removed from the data using Affymetrix Power Tools.

Intensity signals were normalised at exon-level and log$_2$-transformed using the robust multi-array average algorithm in the Affymetrix Expression Console software (build 1.2.1.20). Array hybridisation quality was verified using Affymetrix Expression Console according to the recommendations of the Affymetrix Quality Assessment of Exon and Gene Arrays White Paper. All arrays showed high hybridisation quality and a normal distribution of probe intensity signals.

PCA was performed using Partek GS version 6.6 (Partek Incorporated) and sample outliers removed. Duplicate data for one monozygotic twin pair was processed in both batches to be used as technical replicates and sibling data from the same twin pair within each batch were used as biological replicates. Correlation between replicates was assessed using a Spearman correlation test in R. All replicates showed high correlation ($r^2 > 0.89$). A total of 81 samples from 49 individual twins were included in the analyses.

**Exon array data normalisation and analysis**. Probe sets were summarised to generate gene-level data by calculating the winsorized mean (10 and 90%) using Partek GS. Batch effects were accounted for using the sva ComBat function[70]. Differential gene expression was calculated using Partek GS with a mixed-model analysis of variance (ANOVA) as follows: $Y = \mu + treatment + individual ID + twin ID + PC1 + PC3 + error$. The fitted ANOVA model regressed expression levels at each gene (Y) on fixed-effect terms (treatment, explained by PC2) and on random-effect terms denoting individual ID, family structure and zygosity (twin ID) and PCs explaining most of the data variability (PC1 and PC3).

**SLE patients and healthy controls**. Female patients meeting the American College of Rheumatology (ACR) criteria for the definition of SLE active disease[71] were recruited from Louise Coote Lupus unit, Guy's Hospital ($n = 19$), following informed consent and with ethical approval (Research Ethics Committee; REC 12/LO/1273 and REC 07/H0718/49) and SLE Disease Activity Index (SLEDAI) scores were calculated[72]. The investigator was blinded to SLEDAI scores during measurement of CXORF21 protein abundance. Healthy female controls were recruited from the TwinsUK Bioresource. The TwinsUK study is approved by the research ethics committee at St Thomas Hospital, London. Volunteers gave informed consent and signed an approved consent form prior to participation. Volunteers were supplied with an appropriate detailed information sheet regarding the research project and procedure by post prior to attendance.

**PBMC isolation**. Twenty milliliter of whole blood in EDTA anti-coagulant was taken from female volunteers (SLE or healthy controls). Peripheral blood mononuclear cells (PBMCs) were separated from whole blood using Histopaque-1077 Hybri-Max (Sigma-Aldrich) density centrifugation and plated at 2 × 10^6 cells/ml in RPMI 1640-medium (Gibco) supplemented with 10% foetal calf serum (FCS), 2mM L-glutamine and 100 U/mL penicillin/streptomycin (all from Invitrogen).

**Flow cytometry**. PBMCs were first incubated with Human TrueStain FcX (5 μl; BioLegend) to block Fc receptors, before cell-surface staining with 1 μl anti-human CD14 PerCP-Cy5.5 (eBioscience; 45–0149–42) and 1 μl anti-human CD19 PE (eBioscience;12–0198–42) for 20 min on ice. Cells were fixed with 200 μl 1X stabilising fixative (BD biosciences) and then permeabilized in 0.1% Triton X-100 (Sigma-Aldrich). Fc blocker was added before intracellular staining of 0.1 μg rabbit polyclonal anti-human CXORF21 (Atlas antibodies; HPA001185) or 0.1 μg rabbit monoclonal IgG isotype control (Abcam; ab172730), as appropriate, for 60 min on ice. Following washing, cells were incubated with secondary goat anti-rabbit-Alexa Fluor 488 (Abcam; ab150077) antibody at 1:2000. Cells were washed and resuspended in 250 μl PBS for analysis on BD FACSCanto™ II cytometer (BD Biosciences) using BD FACSDiva software (version 8.0.1; BD Biosciences). Compensation was performed using compensation beads (BD Biosciences), and cytometer settings were standardised using Cytometer Setup and Tracking Beads (BD Biosciences). Following data acquisition, FlowJo v.10.1 software was used to calculate the Median fluorescent intensity (MFI). An unpaired Student's *t*-test was used for case–control analyses. Logistic regression models were fitted for CXORF21 abundance as a function of SLEDAI, and as a function of SLEDAI stratified by age (under/over 35 years of age) with an interaction term. The models were compared using a likelihood ratio test (LRT; d.f. = 5) and BIC using R. Multiple testing was corrected using Bonferroni correction. Preliminary results showed no expression of CXORF21 on cell surface.

**ImageStream analysis**. Multispectral imaging flow cytometry (MIFC) was performed on an ImageStreamX (Amnis) instrument. Golgi colocalisation: 2 × 10^6

cells were fixed with 200 μl 1X stabilising fixative (BD biosciences) and then permeabilized in 0.1% Triton X-100 (Sigma-Aldrich). Fc blocker was added before intracellular staining with 0.1 μg rabbit polyclonal anti-CXORF21 antibody (Atlas antibodies; HPA001185) and secondary goat anti-rabbit Alexa Fluor 488 (Abcam; ab150077) at 1:2000. Cells were then incubated for 60 min on ice with 0.1 μg anti-GM130-Alexa Fluor 647 (Abcam; ab195303). Lysosomal and nuclear colocalisation: $2 \times 10^6$ cells were incubated at 37 °C for 15 min in 1X Assay Buffer and 0.1 μl Lyso-ID Red Detection Reagent and 0.1 μl Hoechst 33342 Nuclear Stain (Lyso-ID Red Detection Kit; Enzo; anti-ENZ-51005–0100). Cells were then fixed with 200 μl 1X stabilising fixative (BD biosciences) and permeabilized in 0.1% Triton X-100 (Sigma-Aldrich). Fc blocker was added before intracellular staining with 0.1 μg rabbit polyclonal anti-CXORF21 antibody (Atlas antibodies; HPA001185) and secondary goat anti-rabbit Alexa Fluor 488 (Abcam; ab150077) at 1:2000. Cells were resuspended in 60 μl PBS. Up to 100,000 images were acquired per sample. Cells were gated on aspect ratio to include only singlets, and the gradient root-mean-square feature to include focused cells. Using the co-localisation mask on the IDEAS software (Amnis), we calculated the overlap of CXORF21 and organelle markers for cellular localisation.

**Immunostaining of autophagic LC3-II and CXORF21.** LCL ($1 \times 10^6$ cells/ml) were starved in EBSS with or without 10 nM or 100 nM bafilomycin A1 (Sigma) for 3 h at 37 °C and 5% $CO_2$ before harvesting, when starvation was required. For LC3 staining, the cells were selectively permeabilized with 0.05% saponin prior to fixation. Cells were fixed in 4% formaldehyde for 20 min at room temperature, then permeabilized with 0.1% Triton X-100 and 2% goat serum (both Sigma-Aldrich) in PBS for 30 min on ice. After overnight incubation in 5% goat serum, cells were Fc receptor blocked (Human TruStain FcX, Biolegend) and incubated with 2 μg/ml rabbit anti-human CXORF21 (Atlas antibodies; HPA001185) and either 2 μg/ml mouse anti-human TLR7 (Novus Biologicals, NBP2–27332) or 40 μg/ml mouse anti-human LC3 (MBL, M152–3) in 5% goat serum for 1 h on ice. Following washing, cells were stained with goat anti-rabbit Alexa Fluor 488 (Abcam; ab150077) and goat anti-mouse Alexa Fluor 594 (Abcam; ab150116), both at 1:2000, in 5% goat serum for 30 min on ice. Cells were washed and mounted in ProLong™ Gold Antifade Mountant containing DAPI (Invitrogen).

**Imaging and analysis.** Imaging was performed at the Nikon Imaging Centre at King's College London. Z stacks were acquired at 0.12 μm step size on an Eclipse Ti-2 Inverted microscope with Vt-iSIM scan head and Hamamatsu Flash4.0 sCMOS camera using a ×100 oil immersion objective. Laser settings, image capture and Richardson-Lucy deconvolution were managed in NIS-Elements. Images were further processed and Pearson's correlation coefficient and Mander's colocalisation coefficient were calculated using the Colocalization Studio plugin[73] in Icy software. Maximum intensity projections are shown for better visualisation. A one-way ANOVA with Tukey multiple comparison correction was performed to test for statistical significance in GraphPad Prism v7.04.

## Data availability

Summary statistics on 10,995 individuals of matched European ancestry (4036 SLE cases, 6959 controls) genotyped on the Illumina HumanOmni1 BeadChip are available at http://insidegen.com/insidegen-LUPUS-data.html. TwinsUK RNASeq data are deposited in European Genome-Phenome Archive (EGAS00001000805). The UK10K (REL-2012–06–02) plus 1000 Genomes Project Phase3 data (release 20131101.v5) merged reference panel (UK10K-1000GP3) was accessed through the European Genome-phenome Archive (EGAD00001000776). All other data are contained within the article and its supplementary information or available upon reasonable request from the corresponding author. The source data underlying Figs. 2d, 6e, f and 7e–g, and Supplementary Figs. 2b, 3, 6b, 12 and 13 are provided as a Source Data file.

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

## Acknowledgements

We thank all volunteers for their contribution to this research. The work was funded by the Medical Research Council L002604/1, Arthritis Research UK (project grants 20265 and 20580, PhD studentship 21252 to S.K.V. and studentship 19983 to C.S.T.D., and clinical fellowship 18239 to M.M.A.F.). K.S.S. receives support from MRC grants MR/R023131/1 and MR/M004442/1. The TwinsUK study was funded by the Wellcome Trust and European Community's Seventh Framework Programme (FP7/2007–2013). The TwinsUK study also receives support from the National Institute for Health Research (NIHR)- funded BioResource, Clinical Research Facility and Biomedical Research Centre based at Guy's and St Thomas' NHS Foundation Trust in partnership with King's College London. The NIHR funded the Biomedical Research Centre (BRC) based at Guy's and St Thomas' NHS Foundation Trust and King's College London. The NIHR funded the BRC Flow Cytometry Core, Guy's Hospital at Guy's and St Thomas' NHS Foundation Trust in partnership with King's College London. We acknowledge Julian Knight and Ben Fairfax for providing X chromosome data on the expression datasets (refs. [42,43]). We thank the Nikon Imaging Centre at King's College London for help with light microscopy.

## Author contributions

C.A.O. performed expression experiments, analysed gene expression and epigenetic data, performed genetic analysis and wrote the manuscript; A.L.R. performed gene expression and Image Stream experiments, performed genetic analysis, analysed data and wrote the manuscript; S.K.V. performed super resolution microscopy experiments and analysed the data; C.S.T.D. performed gene expression experiments in B cells and analysed these data; S.K.V. and C.S.T.D. contributed equally; C.T.B. performed microscopy experiments and analysed the data; A.J.C. designed the microscopy experiments and analysed these data; S.L. performed CXORF21 expression studies in SLE patients; S.J.D. analysed CXORF21 protein expression; L.C. analysed gene expression data; D.L.M. analysed genetic data; L.L. J. validated the anti-CXORF21 antibody; L.B. performed gene expression studies in resting and IFN-stimulated B cells; A.Z. and K.S.S. analysed X chromosome skewing and expression of *CXorf21* in TwinsUK; M.M.A.F. designed and supervised the B cell expression studies; D.S.C.G. designed and supervised the study; T.J.V. designed and supervised the study and contributed to writing the manuscript.

## Additional information

**Competing interests:** The authors declare no competing interests.

