## [Peer Review File · Nature Communications]

Reviewer #1 (Remarks to the Author):

Mechanism of sexual dimorphism observed in several autoimmune disease, including SLE, has not fully understood. The manuscript "Altered expression of the interferon inducible X-linked gene, CXorf21, contributes to sexual dimorphism in Systemic Lupus Erythematosus" by Odham, C.A. et al investigate the CXorf21 as a potential causal factor of SLE. Authors made several interesting and scientifically important observations. i) Correlation between expression of CXorf21 and SLE, ii) polymorphism which is highly associated with SLE can affect expression of CXorf21 expression and iii) level of the CXorf21 is positively correlated with SLE disease activity and discrete expression in innate (monocyte and neutrophil) and adaptive (B cells) in lymphoid cells. Author tried to prove the scientific observation by using genetic, epigenetic and cellular experiments. Identification of new gene which potentially results in sexual dimorphism and causal factor to contribute to SLE development is interesting. Most of experiments, data analysis, and conclusions drawn from the primary data is well designed and accepted. There are couple of points which might improve quality of study if corrected.

Specific points:

1. In page 8, author described the exclusion of samples due to the variability of X-chromosome expression. Description and knowledge how to stratify the sample collection, what is the baseline for inclusion or exclusion of samples? How author set the <0.2 or >0.8 ? Samples variation is often observed in human study, but not all made this selection.
2. Is Figure 1 C adopted from public data base or author's own data? If this is public data, it is better move to supplementary data. In fact, many data analysis (genetic and epigenetic data) are not clearly described whether they are made by author's own analysis or from public data.
3. To support the data figure 1C, author can try to confirm the binding of TFs to the promoter of CXorf21 by EMSA or ChIP assay.
4. Anti-CXorf21 antibodies (from Atlas antibodies) are rabbit polyclonal antibodies, therefore, proper control should be rabbit polyclonal antibodies not rabbit anti-human IgG. Also, in the company product data show significant expression of CXorf21 in the cell lines, RT-4 (epithelial cells from urinary bladder), U-251 MG (human brain), and A-431 (skin epidermis). However, author described that expression of CXorf21 is epigenetically inert in non-immune cells (page 12, line 295-297). Either author should correct the conclusion or re-consider about the quality of antibody specificity.
5. Figure 6, co-localization of LC3 and CXorf21 is not convincing. This is from a single cell. Author should show lower power images including many cells and also quantification of the co-localization by imagestream (shown in supplementary figure 9 and 10).
6. Page 16, in Cis-eQTL association analysis, "unpaired t-test" is right statistical method?

Reviewer #2 (Remarks to the Author):

Odhams et al. present a manuscript aiming to present molecular and genetic mechanisms that lead to the female predominance of Systemic Lupus Erythematosus (SLE).

Major:

1. Figure 1: The results are robust, but I am not convinced that the experiment is optimally designed given the premise of this study. TLR7 and TLR9 are induced in activated cells. If CXORF21 is operating downstream of these TLRs, it would be more reasonable to pre-stimulate the cells (e.g. with IFN-beta or maybe LPS), confirm upregulation of TLR7 and TLR9, and then perform the experiment downstream of stimuli that are specific to TLR7 and TLR9. The very TFs that the authors suggest are driving inflammatory-mediated expression are IRFs and STATs that can be activated through these receptors.

Similarly, the conclusions in Sup Fig 7 might change if the cells were pre-stimulated to have TLR7 and TLR9 expression before appropriate stimulation through these endosomal pattern recognition receptors.

This criticism is especially relevant given the data presented by the authors in Figure 5, that the SLE activity score (which is associated with Type I IFN activity, and thus immune cells primed to express more TLR7 and TLR9) is associated with increased CXORF21 expression.

2. Given the literature describing the X-chromosome encoded and sex-dependent TLR7 expression and activity in immune responses, the study needs further investigation into the relationship between TLR7 and CXORF21 expression. In addition to the first major point, Are the two genes co-expressed? Does this co-expression change in the context of the SLE-risk alleles? Is the co-expression or genotype-dependent expression sex-dependent?

3. The type of "epigenetic fine mapping" presented is fairly non-standard. If true, then one would expect genotype-dependent chromatin marks in females that are heterozygous for the variant rs887369. This analysis is needed to collaborate the conclusions made based on the position of this variant relative to the position of the ChIP reads from anti-activated histone marks.

4. For HiC experiments analyzed, were these performed in male or female cell lines and what was the genotype? Were any done in females with a heterogeneous genotype at rs887369?

Minor

1. Supplementary Figure 4:
 - a. For the A panel: What is the statistical significance for the sex-dependent expression after multiple testing corrections?
 - b. For panel D: This is a confusing figure to interpret. What do the "expression values" mean? Are the FPKMs? Do we interpret these to mean that the "significant" male-female difference in monocytes is 8.0 vs 8.3 is a ~3% difference? That is very different than the >3-fold difference seen in the LCLs. What multiple testing threshold was used to set "significance" for sex-dependent differences for panel D?
 - c. The title of Sup Fig 4 seems to conflate the expression differences in LCLs and primary cells - this needs to be better justified.
2. On lines 109 and 110, The authors state that "Interestingly, CXORF21 was also found to interact with itself, suggesting probable oligomerization of this protein". Is there data or a reference to support this conclusion? Does the oligomerization happen in an SLE or sex-dependent manner?
3. In Supplementary Figure 1, RNA-seq data from the Human Protein Atlas is used to provide expression of the eight candidate genes across various tissues. Can the authors also provide protein data from the same source? Is this data consistent with GTEx (from which genotype-dependent expression could also be assessed)?
4. Supplementary Figure 2: Are the samples from which the epigenetic marks assessed male or female? Is data available such that matched male and female epigenetic marks could be shown? For example, a more limited set of marks could be shown from male and female 1000 genomes cell lines.
5. In Figure 1 and lines 142-145, are the authors concluding that there is only 1 ISRE within 1 MB of the tag SNP? 25 bp is an extremely narrow analysis of a promoter region (promoter regions have different definitions, but when defined by distance from a TSS are generally defined as 1kb, 2kb, or 5kb upstream of the TSS). Also, is there only one ISRE in this promoter (with an extended definition)? Similarly, I would zoom out and show 5 kb 5' of the TSS in Supplementary Figure 5 to match the 5kb 3' region shown.
6. If only males were used in parts of Figure 2, that needs to be indicated in the Figure legend. If not, then please separate male and female LCLs in the eQTL analysis.

7. On line 184, I would provide R2 and not D' - this is likely to further enforce the point that there is no significant LD between this variant and nearby haplotypes.

8. Throughout the manuscript, an effort should be made to identify the threshold for significance in the context of multiple testing. These thresholds should be clearly defined in figure legends and (when appropriate) the methods section.

Reviewer #3 (Remarks to the Author):

This paper describes investigations to identify the gene and mechanism underlying an established association to SLE risk on chromosome X. They identify CXorf21 as the most likely candidate on the basis of expression patterns of this gene which is more concentrated in immune cells than neighbouring genes. They go on to show that expression of CXorf21 differs between males and females, and between activated vs non-activated immune cells, and is affected by XCI. The link to SLE is through the SLE risk SNP which is an eQTL for CXorf21 expression. On this basis, this gene is claimed as part of sex imbalance in SLE (in the title) and XCI more widely in the discussion.

This is an interesting paper. It is refreshing to read the story of investigations to identify a disease risk gene and mechanism, and the theory of variable XCI contributing to sexual dimorphism in autoimmune disease risk is also interesting. Especially, this is a big positive in contrast to the trend in big papers towards hypothesis free analysis of ever larger datasets!

However, the evidence presented is not convincing for all claims. It is rather a collection of circumstantial pieces of evidence, and as a reader we are not told what was sifted out. RNA eQTL signals are weak, and found only in subsets of datasets (eg only in males). The protein work is much more convincing. Many p values are of the order 0.001 - 0.1 which, given the number of pieces of evidence presented (and the unknown number of pieces not presented) prevents strong interpretation because of multiple testing issues. Claims to “nominal significance” and “show a trend” do not help.

To strengthen the work, there needs to be more complete summary of negative evidence, and a formal synthesis of evidence presented. Even for studying sexual dimorphism in expression, it is hard to keep track of the cell types and activation states, fold changes and p values should be presented for each combination considered and summarised more clearly. More care needs to be taken to present a readable and engaging story. It is not an easy paper to write, but currently it is also not an

easy paper to read, with a series of results presented in each section, and the onus falls on the authors to make it more readable.

Major comments:

The title is too strong - that CXorf21 contributes to sexual dimorphism in SLE is a hypothesis, not something shown. Similarly Fig 7 needs to be presented as a working hypothesis, no more.

The abstract claims to interaction between sex-specific and IFN-inducible expression, but not a single test for interaction is performed in the paper. The word interaction should be removed.

Page 4 Line 70 Lead SNP is apparently in UTR. (Assume from previous sentence the lead SNP is rs887369?) However looking in ensembl or dbsnp, or Fig 7, it appears to be a synonymous SNP in protein position 209 out of 301 in the protein. This also affected the interpretation of data on H3K36me3. It is known that H3K36me3 can 'build up' at 3' end of genes undergoing transcription. Is this then of any interest/relevance in selecting a causal SNP?

Fine mapping of causal variant as rs887369 is also not convincing. Figure 3C shows gaps in SNP coverage which are not present looking at common snps on eg UCSC genome browser. How were these SNPs selected? It would seem sensible to focus on fine mapping first, then eQTL analysis, then present the characterisation of CXorf21 - this would lead from standard analyses to more interesting/exciting, rather than selecting a gene of interest because it matches an XCI hypothesis and then amassing evidence in support. Was rs887369 the strongest SNP in the eQTL analyses?

Details are missing for many of the statistical analyses conducted. Eg how was haplotype analysis conducted - all male/female? What is the number of subjects? How were female hets/homs coded? How did they conduct the enrichment analysis for epigenetic marks and why H3K36me3, what are the results from the other 12 histone marks analysed?

The link of CXorf21 is via correlation with SLEDAI. But there is an obvious outlier in the plots. What happens to the p-value with outlier removed? Is this tested by Pearson's correlation? A more robust test for small, non-normal data, would be to use Spearman's.

Minor comments:

Fig 1 3d plots are very hard to read - a 2d representation showing the gradients between each pair of 4 bars of points would be much clearer.

Generally, page 5-6, difficult to know exactly what is the size of effect, and key differences between cells. Giving estimated fold changes as well as direction and p value would be helpful.

Supplementary table 2 - GPR65 (2nd in table) that is coexpressed with CXorf21 is in risk locus for Crohn's, UC, MS and Ankylosing spondylitis is this of any interest/relevance ?

If 5 SNPs are "perfectly correlated" they should all have identical p values for all tests. Why do they not? And how does this enable them to focus exclusively on rs887369 before looking at H3K36me3 (ignoring that H3K36me3 does not convincingly fine map any causal variant)?

Line 208 - H3K36Me3 does not get "bound" - this is a covalent histone modification modulated by for example SET2.

L219 It is good scientific practice to credit those who have created data/resources you then use. Please give url/reference for "ChIP resource" and the data it presents.

Line 173 - Text n=412 plot says 414 - typo?

Figure 7 should TTS be TSS ? (typo?)

Dear Editor,

Please find below our point-by-point response to the reviewer's comments.

Reviewer #1

1. *In page 8, author described the exclusion of samples due to the variability of X-chromosome expression. Description and knowledge how to stratify the sample collection, what is the baseline for inclusion or exclusion of samples? How author set the <0.2 or >0.8? Samples variation is often observed in human study, but not all made this selection.*

The reviewer makes a good point here and we seek to answer the point by now providing a more detailed description of the analyses of the LCL samples from the TwinsUK cohort, which were used to evaluate *cis* eQTL in females for *CXorf21*. There is more description of the analysis in the Results section itself (pages 6-7) and we provide more details in the Methods section (pages 17-18, and quoted below), including references from leading investigators in the field of X chromosome genetics, to support the numerical criteria that we used for exclusion for eQTL analysis on the basis of extreme X chromosome skewing. Here is an extract from the amended methods section:

“Individuals were firstly assessed for skewed X-chromosome inactivation patterns using allele-specific expression of *XIST* to estimate the proportion of X inactivation from each parental X chromosome. Individuals were removed if the allele-specific expression of *XIST*-linked SNPs was < 0.2 or > 0.8, these parameters were chosen on the basis of precedence (References below). *cis*-eQTL analysis in the twins was performed as above of rs887369 against residualized expression data, exon count residuals corrected for probabilistic estimation of expression residuals (PEER) factors and family relatedness.”

A. K. Naumova, R. M. Plenge, L. M. Bird, M. Leppert, K. Morgan, H. F. Willard, and C. Sapienza. Heritability of X chromosome--inactivation phenotype in a large family. *Am J Hum Genet.* 1996 Jun; 58(6): 1111–1119.

Amos-Landgraf JM1, Cottle A, Plenge RM, Friez M, Schwartz CE, Longshore J, Willard HF. X chromosome-inactivation patterns of 1,005 phenotypically unaffected females. *Am J Hum Genet.* 2006 Sep;79(3):493-9.

Kristiansen M1, Knudsen GP, Bathum L, Naumova AK, Sørensen TI, Brix TH, Svendsen AJ, Christensen K, Kyvik KO, Ørstavik KH. Twin study of genetic and aging effects on X chromosome inactivation. *Eur J Hum Genet.* 2005 May;13(5):599-606.

2. Is Figure 1 C adopted from public data base or author's own data? If this is public data, it is better move to supplementary data. In fact, many data analysis (genetic and epigenetic data) are not clearly described whether they are made by author's own analysis or from public data.

We thank the reviewer for highlighting the need for clarity in presenting the datasets used. We have included an additional display item (Supplementary Figure 1) showing an overview of all data used, whether public or in-house. We hope the reviewer agrees that this addition aids the interpretation of the manuscript. Additionally, each figure legend clearly states whether the source was public or in-house data. Finally, to specifically answer the question; Fig 1C (now Fig 5B) was generated from ENCODE data.

3. To support the data figure 1C, author can try to confirm the binding of TFs to the promoter of CXorf21 by EMSA or ChIP assay.

We agree that we could confirm binding to the promoter of *CXorf21* with additional laboratory experiments. However, the SLE association signal that we described is located at the 3' end of the gene. We have interrogated two datasets, RoadMap and BLUEPRINT and both of these datasets give evidence for the binding of anti-H3K36me3. The situation in the region of the promoter of the gene is much more complex, with ENCODE providing evidence for the binding of multiple different TFs in LCLs. A detailed exploration of the binding by ChIP and/or EMSA in a variety of different *ex vivo* immune cells would be a substantial volume of work that would represent a component of another manuscript describing the transcriptional regulation of this gene in different immune cells.

4. Anti-CXorf21 antibodies (from Atlas antibodies) are rabbit polyclonal antibodies, therefore, proper control should be rabbit polyclonal antibodies not rabbit anti-human IgG. Also, in the company product data show significant expression of CXorf21 in the cell lines, RT-4 (epithelial cells from urinary bladder), U-251 MG (human brain), and A-431 (skin epidermis).

However, author described that expression of CXorf21 is epigenetically inert in non-immune cells (page 12, line 295-297). Either author should correct the conclusion or re-consider about the quality of antibody specificity.

Firstly, it is important to reiterate that we validated the specificity of the anti-CXORF21 antibody (Supplementary Figure 3) following RNA-knock down of *CXorf21* in LCLs. Secondly, we thank the reviewer for spotting an error in the reporting of our methods; the isotype control used was rabbit monoclonal IgG (Abcam), a non-specific control antibody which provides information on background staining. This antibody was used for the flow cytometry data, where conclusions are drawn between cases and controls (Supplementary Fig. 12), and across cases (Supplementary Fig. 13). We can assume that there would be no difference in the amount of non-specific binding of between individuals, therefore the isotype control would not introduce a bias into the experiment.

Reviewer #1 also draws attention to the datasheet of the Atlas Antibodies rabbit anti-CXorf21 showing variable binding of the antibody in three cell lines. The three cell lines in question are 1) transitional cell papilloma aneuploidy RT-4 male cells (very weak signal); 2) glioblastoma astrocytoma U-251 MG male cells (weak signal); 3) squamous carcinoma hypertriploid A-431 female cells (strong signal). As all three are tumour cell lines, we do not believe they are necessarily physiologically representative of CXORF21 expression in immune cell types. Therefore, this fact should not alter our conclusions, drawn from protein expression data in non-cancerous cell types, that CXORF21 is inert in non-immune cell types. It is interesting that the only strong signal seen here is in the triploid cell line originating from a female (A-431).

Finally, we should also mention that we examined the expression of CXorf21 across a whole range of human tissues from the Human Protein Atlas. The results (Supplementary Figure 7) confirm that on ranking, the tissues with the highest levels of expression are all immune related (namely lymph node, bone marrow, appendix, spleen and tonsil). We accept that in this dataset there is evidence for some expression in the lung and urinary bladder, but the source of this expression may be immune cells in these tissues.

5. Figure 6, co-localization of LC3 and CXorf21 is not convincing. This is from a single cell. Author should show lower power images including many cells and also quantification of the co-localization by imagestream (shown in supplementary figure 9 and 10).

The reviewer makes an important and highly relevant point here. The staining with LC3 was relatively weak and this is to be expected in LCLs, which have limited amount of cytoplasm and require stimulation and blockade of autophagy to reveal autophagosomes. In order to gain more certainty here we have repeated these experiments using Structured Illuminated Microscopy (SIM) and included an analysis of *ex vivo* B cells (revised Fig. 7). We consistently observe relatively weak LC3 staining and thus we have tempered our conclusions about colocalization of CXORF21 with LC3 in the revised manuscript. In addition, the microscopy studies have been extended by included staining with anti-TLR7, which we think provides more robust results (see response to reviewer 2 below) and are shown in revised Fig. 6.

6. Page 16, in *Cis*-eQTL association analysis, “unpaired *t*-test” is right statistical method?

In the *cis*-eQTL analyses in *ex vivo* cells, it is apparent that the distributions of the data are non-skewed and that there are reasonable numbers ($n > 30$) within each group (Fig 2E). These points justify the use of an unpaired *t* test.

Reviewer #2 Major:

1. Figure 1: The results are robust, but I am not convinced that the experiment is optimally designed given the premise of this study. TLR7 and TLR9 are induced in activated cells. If CXORF21 is operating downstream of these TLRs, it would be more reasonable to pre-stimulate the cells (e.g. with IFN-beta or maybe LPS), confirm upregulation of TLR7 and TLR9, and then perform the experiment down stream of stimuli that are specific to TLR7 and TLR9. The very TFs that the authors suggest are driving inflammatory-mediated expression are IRFs and STATs that can be activated through these receptors.

Similarly, the conclusions in Sup Fig 7 might change if the cells were pre-stimulated to have TLR7 and TLR9 expression before appropriate stimulation through these endosomal pattern recognition receptors.

This criticism is especially relevant given the data presented by the authors in Figure 5, that the SLE activity score (which is associated with Type I IFN activity, and thus immune cells

primed to express more TLR7 and TLR9) is associated with increased CXORF21 expression.

The reviewer makes some interesting points here, but we do not understand on what evidence they suggest that CXorf21 operates 'downstream of these TLRs'. We stimulated the B cell with interferon as we had data from *ex vivo* cells showing that interferon elevated expression of CXORF21 and is known to be an important cytokine in SLE pathogenesis. In view of the data shown in Supplementary Table 6 in which we show that there is co-expression of CXorf21 and TLR7, which has been replicated in an RNA-Seq dataset in the revised manuscript, we sought to determine whether there was a possible functional interaction between CXORF21 and TLR7. We undertook this analysis by using Structured Illumination Microscopy (SIM) and we have shown that there is some colocalisation of CXorf21 and TLR7 (revised Figure 6). We do not know in which cellular compartment(s) this takes place. There are clearly many further questions that can be asked about the functional effects of CXORF21 and its potential role in the intracellular TLR pathways. However, a detailed investigation of the functional effects of CXORF21 would justify another full manuscript.

2. Given the literature describing the X-chromosome encoded and sex-dependent TLR7 expression and activity in immune responses, the study needs further investigation into the relationship between TLR7 and CXORF21 expression. In addition to the first major point, Are the two genes co-expressed? Does this co-expression change in the context of the SLE-risk alleles? Is the co-expression or genotype-dependent expression sex-dependent?

We have investigated the co-expression of CXORF21 and TLR7 and these results are displayed in Supplementary Table 6. We have since replicated this finding in an independent RNA-Seq dataset from female LCLs (see page 11 of manuscript - $n = 271$; $\rho = 0.38$; $P = 6 \times 10^{-11}$). We do not have access to genotype data on all the samples that were employed in these analyses and hence cannot answer the reviewer's question using these datasets. We did study smaller datasets from *ex vivo* cells as shown in Fig 2. However, it is important to point out that when considering the effect of genotype on gene expression in females as suggested by the reviewer, that the confounding effect of X chromosome skewing will be a major confounding factor on the results unless the degree of skewed X chromosome expression is accounted for as we have done (see response to first point of reviewer 1).

3. *The type of "epigenetic fine mapping" presented is fairly non-standard. If true, then one would expect genotype-dependent chromatin marks in females that are heterozygous for the variant rs887369. This analysis is needed to collaborate the conclusions made based on the position of this variant relative to the position of the ChIP reads from anti-activated histone marks.*

We have been careful in the revision of the manuscript to avoid over emphasis of fine mapping of the risk haplotype. The epigenetic data show elevation of epigenetic signal of histone modification towards the end of the associated haplotype that overlaps with *CXorf21* transcription, however, we are careful now to avoid the implication that we have defined the risk allele within this haplotype and we accept that the functional effect may arise from more than one of the variants on the risk haplotype.

We agree with the reviewer that looking at sex-specific effects using epigenetic data would be potentially interesting. Our sources of epigenetic data were from the RoadMap and BLUEPRINT consortia. Results from RoadMap derived from a very limited number of individuals and the results are combined. Thus, we cannot use data to address sex specific effects as suggested by the reviewer. The BLUEPRINT data comprise more individuals than RoadMap and we examined this dataset for sex-specific differences in annotation across the risk haplotype. These results were inconclusive, however as we think this is an important question we have included the results in the revision as Supplementary Figure 4 and Supplementary Table 2.

4. *For HiC experiments analyzed, were these performed in male or female cell lines and what was the genotype? Were any done in females with a heterogeneous genotype at rs887369?*

We agree that this is an important question; however, the original promoter capture dataset (Javierre *et al* 2016 - [https://www.cell.com/cell/fulltext/S0092-8674\(16\)31322-8](https://www.cell.com/cell/fulltext/S0092-8674(16)31322-8)) does not specify the sex or the sample genotypes of the 17 human primary hematopoietic cell types analysed. In fact, according to the publication methods, many of the experiments for each specific cell type are pooled from multiple healthy donors – making it not possible to deconvolute the sex or genotype of the individual donors. It is important to note that this analysis was not performed in cell lines, but rather in primary venous or cord blood and *in vitro* cultured and differentiated cell types from healthy donors. The methods section (Promoter capture Hi-C chromatin interaction data), results (The SLE-associated haplotype

interacts with the promoter of *CXorf21*), and Figure legend (Fig 3C) have been amended to clarify the above points. As with the investigation of genotype-dependent epigenetic modification around the 3'-UTR of *CXorf21*, we also agree that genotype-dependent chromatin interaction is an interesting line of experimentation. Such experiments will help disentangle the molecular link between genetic alteration and increased *CXorf21* expression in SLE.

Reviewer #2 Minor:

1. A. *Supplementary Figure 4 panel A: What is the statistical significance for the sex-dependent expression after multiple testing corrections?*

We have now included a formal sex-dependent fold-change expression analysis of *CXorf21* across all GTEx cell/tissue types where both male and female samples were available (45 different cell/tissue types). RNA-Seq expression data (Transcripts Per Million – TPM) were downloaded from GTEx along with sample phenotype (sex) and experiment ID (cell/tissue source per sample). A *t*-test was performed between males and female samples using the TPM values and grouping by cell/tissue type. Following this analysis, we found that of the 45 cell/tissue types assessed, only EBV-transformed lymphocytes and thyroid tissue yielded significant *P*-values (1.10×10^{-05} and 2.65×10^{-03} , respectively), using a Bonferroni multiple testing correct at a cut-off of $P_{BF} < 0.05$. We now provide the full results of this analysis in Supplementary Table 3 and have written a more detailed methods (page 18; *Genotype-expression cohorts and cis-eQTL association analysis*) and results (page 9; *Sexual dimorphic expression of CXorf21 is magnified in activated immune cells*) section to reflect this inclusion.

1. B. *Supplementary Figure 4 panel D: This is a confusing figure to interpret. What do the "expression values" mean? Are the FPKMs? Do we interpret these to mean that the "significant" male-female difference in monocytes is 8.0 vs 8.3 is a ~3% difference? That is very different than the >3-fold difference seen in the LCLs. What multiple testing threshold was use to set "significance" for sex-dependent differences for panel D?*

The data presented in Figure 4D are derived from the Fairfax *et al* 2014 *Science* publication where gene expression profiling was measured by microarray using the Illumina HumanHT-12 v4 BeadChip gene expression array platform. Expression values therefore correspond to the \log_2 normalized fluorescence intensity values and are thus unitless relative values. In this

sense, the fold-change of these microarray experiments can be calculated as $2^{(\text{female expression} - \text{male expression})}$ = fold-change i.e. for LPS stimulated monocytes at 24 hours: $2^{(8.224915 - 7.960083)}$ = 1.2 fold-change (20% greater in females) which is highly significant when comparing the differences between the two groups using an un-paired *t*-test ($P=1.41 \times 10^{-12}$). Again, with the above point, we corrected for multiple tests using the conservative Bonferroni method at a cut-off of $P_{BF} < 0.05$. We have now updated the microarray figures to annotate the y-axis with “log₂ scale” and have also provided a new table (Supplementary Table 4) with the full results of the differential expression analysis between males and females using the microarray data in primary immune cell types. The methods section (page 18; *Genotype-expression cohorts and cis-eQTL association analysis*) has also been updated to reflect these changes.

1. C. The title of Sup Fig 4 seems to conflate the expression differences in LCLs and primary cells - this needs to be better justified.

We thank the reviewer for this comment. We have now annotated the y-axes of the graphs in revised Supplementary Figure 2 with the cell types under investigation. We have also moved the results of the primary *ex vivo* expression data to another separate table (Supplementary Table 4) to avoid confusion. Additionally, the title of Supplementary Fig 4 (now Supplementary Figure 2) is: ‘Cis-eQTL analysis of SLE associated SNP, rs887369, on gene expression across an array of immune cell-types in females only, with qPCR validation’. In the figure legend, we now underline the cell types used in each experiment.

2. On lines 109 and 110, The authors state that "Interestingly, CXORF21 was also found to interact with itself, suggesting probable oligomerization of this protein". Is there data or a reference to support this conclusion? Does the oligomerization happen in an SLE or sex-dependent manner?

We thank Reviewer 2 for highlighting that this statement was not clearly written. We were referring to the BioPlex data. This has been amended (see page 12) to read: “Interestingly, in the BioPlex data, CXORF21 was also found to interact with itself, suggesting probable oligomerization of this protein.”

3. In Supplementary Figure 1, RNA-Seq data from the Human Protein Atlas is used to provide expression of the eight candidate genes across various tissues. Can the authors also provide protein data from the same source? Is this data consistent with GTEx (from which genotype-dependent expression could also be assessed)?

This is an excellent point. In addition to the HPA RNA-Seq data from primary cell types, we have now included RNA-Seq data of *CXorf21* across multiple different tissue and cell types from: GTEx, FANTOM5, and from the cell-line study of HPA (across 12 cell-types groups). These additional data strongly support our original findings that *CXorf21* expression at the RNA level is greatest in immune cell-types. For example, the HPA, GTEx, and FANTOM5 datasets all show that *CXorf21* expression is highest in the appendix and spleen. HPA cell-line data in fact show completely restricted expression in myeloid and lymphoid cell-lines. We have now presented these data in Supplementary Figure 8.

Using the GTEx data, we were unable to find a significant genotype-dependent expression effect (eQTL) for the lead SNP rs887369. This was expected however as the data are from a combination of males and females, and the eQTL model has not been adjusted for the degree of X-skewing (*CXorf21* is known to escape X-inactivation). We found that rs887369 is a significant *cis*-eQTL for *CXorf21* in multiple cell types (both in cell-lines and in primary *ex vivo* immune cells) when stratifying for sex and for degree of X-chromosome skew (page 6; *The rs887369 risk haplotype increases the expression of CXorf21 in male and female LCLs*).

We have now included the protein data from the HPA to strengthen our argument that *CXorf21* is expressed highly in immune cell types (Supplementary Figure 11). Though we find CXORF21 protein abundance greatest in the immune tissue types: thyroid, bone marrow, tonsil, and appendix – which supports the RNA data – we also find CXORF21 protein ‘equally’ abundant in ‘non-immune’ tissue types such as the stomach, placenta, and skin. CXORF21 protein abundance is found to be ‘medium’ in these tissue types. We have adjusted our manuscript to state that CXORF21 expression at the protein level largely reflects the RNA data but is not as compartmentalised and potentially more ubiquitous across tissue groups.

4. *Supplementary Figure 2: Are the samples from which the epigenetic marks assessed male or female? Is data available such that matched male and female epigenetic marks could be shown? For example, a more limited set of marks could be shown from male and female 1000 genomes cell lines.*

We were unable to make robust conclusions on differential H3K36me3 signal between the sexes in BLUEPRINT data as the sample sizes per cell-type were too small and the variability in fold-enrichment within these samples was too great. See page 8 of manuscript: “We were unable to make robust conclusions on differential H3K36me3 signal between the sexes as the sample sizes per cell-type were too small (Supplementary Fig. 4, Supplementary Table 2).”

5. In Figure 1 and lines 142-145, are the authors concluding that there is only 1 ISRE within 1 MB of the tag SNP? 25 bp is an extremely narrow analysis of a promoter region (promoter regions have different definitions, but when defined by distance from a TSS are generally defined as 1kb, 2kb, or 5kb upstream of the TSS). Also, is there only one ISRE in this promoter (with an extended definition)? Similarly, I would zoom out and show 5 kb 5' of the TSS in Supplementary Figure 5 to match the 5kb 3' region shown.

We assayed for ISREs within the gene locus of *CXorf21* and 5kb upstream of the TSS. We found one ISRE within this region which was located +25bp upstream of the immediate *CXorf21* promoter. This promoter ISRE was heavily bound by NF-KB, STAT1, STAT2, STAT3, IRF4, and IRF3 using ENCODE data in LCLs (revised Figure 5B). The nearest additional ISRE to this one was located 70kb downstream of the 3'-end of *CXorf21*. Therefore, we conclude that there is only one promoter bound ISRE for *CXorf21*. We have included these details in the appropriate results section. We thank the reviewer for this advice and have amended the graph to show the entire 3' and 5' span of *CXorf21*.

6. If only males were used in parts of Figure 2, that needs to be indicated in the Figure legend. If not, then please separate male and female LCLs in the eQTL analysis.

All figure legends have now been revised, making them clearer with regards to the data used. They have also been reordered following the re-write of the manuscript. To answer this particular point, please see Fig 2 and its legend.

7. On line 184, I would provide R2 and not D' - this is likely to further enforce the point that there is no significant LD between this variant and nearby haplotypes.

Thank you for the suggestion. We have included the R2 value here along with the D', which demonstrates the weak LD between the associated haplotype and neighbouring haplotypes. Please see page 5: "The associated haplotype is distinctly separated from neighbouring haplotypes by high recombination ($D' < 0.6$, $r^2 < 0.2$)"

8. *Throughout the manuscript, an effort should be made to identify the threshold for significance in the context of multiple testing. These thresholds should be clearly defined in figure legends and (when appropriate) the methods section.*

We thank Reviewer #2 for drawing our attention to this. In the re-write of the manuscript we have made every effort to clearly state significance thresholds in each test.

Reviewer #3 Major comments:

1. *The title is too strong - that CXorf21 contributes to sexual dimorphism in SLE is a hypothesis, not something shown. Similarly Fig 7 needs to be presented as a working hypothesis, no more.*

We have edited both the title of the manuscript and revised Fig. 8 (previously Fig. 7) to better reflect this being a hypothesis. In addition, reference to rs887369 has now been replaced with 'risk haplotype' in Fig. 8 and throughout the manuscript.

2. *The abstract claims to interaction between sex-specific and IFN-inducible expression, but not a single test for interaction is performed in the paper. The word interaction should be removed.*

We have removed the word interaction from the Abstract as suggested by the reviewer.

3. *Page 4 Line 70 Lead SNP is apparently in UTR. (Assume from previous sentence the lead SNP is rs887369?) However looking in ensembl or dbsnp, or Fig 7, it appears to be a synonymous SNP in protein position 209 out of 301 in the protein. This also affected the interpretation of data on H3K36me3. It is known that H3K36me3 can 'build up' at 3' end of genes undergoing transcription. Is this then of any interest/relevance in selecting a causal SNP?*

We have modified the manuscript throughout to place less emphasis on the causal role for the rs887369 SNP and more emphasis on the association of the haplotype. We agree that the functional effect of the associated haplotype may stem from more than one polymorphism and our manuscript does not include targeted experiments to isolate the actual causal variant(s) on the risk haplotype.

4. Fine mapping of causal variant as rs887369 is also not convincing. Figure 3C shows gaps in SNP coverage which are not present looking at common snps on eg UCSC genome browser. How were these SNPs selected? It would seem sensible to focus on fine mapping first, then eQTL analysis, then present the characterisation of CXorf21 - this would lead from standard analyses to more interesting/exciting, rather than selecting a gene of interest because it matches an XCI hypothesis and then amassing evidence in support. Was rs887369 the strongest SNP in the eQTL analyses?

We agree with Reviewer #3 that the reordering of the manuscript to display the fine-mapping first. We believe this has created a more engaging narrative for the reader. As stated above in the previous response, we have reduced the emphasis in the revision on the causality of one particular variant. In response to the specific questions of the reviewer, rs887369 was indeed the strongest SNP in the eQTL analysis, as shown in Fig 2.

However, we disagree with the reviewer's comment suggesting the variation presented on the UCSC browser suggests "gaps in SNP coverage" in our fine-mapping resulting in the data being unconvincing. UCSC displays genetic variation captured by multiple large-scale publicly available whole-genome sequencing studies. Whereas our data were generated from genotype imputation (using the merged UK10K and 1000 Genomes project). A discrepancy between the variants captured by the UCSC and our imputation based fine-mapping – as would be seen for any study - is therefore not only completely legitimate but also to be expected.

In summary, and as described in full in Methods (see page 17) we used genotype data from the Illumina HumanOmni1 BeadChip and imputed genotypes to the density of the UK10K-1000GP3 reference panel across the X:30077468-31077846 1Mb region (24,700 variants are captured by the reference panel in this region). Any SNPs represented in Fig1C (our fine-mapping of the region), will have passed the following sequential QC steps

1. Imputation info score >0.5
2. Missingness <10% across all 10,995 samples
3. Minor allele frequency > 1%,
4. HWE > 1×10^{-4}
5. Mapping to 100Kb window around the association

The final genotyping dataset comprised 657 SNPs, as stated on page 18 of the manuscript.

5. Details are missing for many of the statistical analyses conducted. Eg how was haplotype analysis conducted - all male/female? What is the number of subjects? How were female hets/homs coded? How did they conduct the enrichment analysis for epigenetic marks and why H3K36me3, what are the results from the other 12 histone marks analysed?

We agree with the reviewer that the original manuscript lacked some clarity in this point. The full methods used for the haplotype analysis have now been included (see pages 17-18). Briefly, genotype data from 10,995 individuals were analysed, implementing X chromosome analysis in Haploview 4.2. Haplotype blocks across a 100kb region anchored on rs887369 were defined by the confidence intervals algorithm (*Gabriel et al*) and SNP and haplotype association testing performed by Chi-Square test using marker thresholds of MAF > 0.01 and HWE > 1×10^{-4} , and minimum genotype rate > 90% (657 SNPs total). As for enrichment analyses, we took all marks across all encode cell types and only H3K36me3 was found to be enriched,

6. The link of CXorf21 is via correlation with SLEDAI. But there is an obvious outlier in the plots. What happens to the p-value with outlier removed ? Is this tested by Pearson's correlation? A more robust test for small, non-normal data, would be to use Spearman's.

We thank the reviewer for this comment, which has led to more robust analysis of the correlation between CXORF21 protein abundance and SLEDAI in a larger cohort size. We show that there is a positive correlation between CXORF21 and SLEDAI in age-stratified females. We fitted two regression models: 1) CXORF21 as a function of SLEDAI; 2) CXORF21 as a function of SLEDAI stratified by age (under/above 35 years of age; Supplementary figure 13) with an interaction term. We tested the robustness of the SLEDAI*Age interaction model compared to that of SLEDAI as a single variable using both a likelihood ratio test and BIC, both of which supported the rejection of the single variable model in favour of the interaction

model for both monocytes and B cells (see page 12 of manuscript). We see a significant interaction term in the monocyte data ($P= 0.002$), though the interaction term in B cells does not pass Bonferroni correction ($P=0.01$). We discuss the relevance of these results on page 16.

Reviewer #3 Minor comments:

1. Fig 1 3d plots are very hard to read - a 2d representation showing the gradients between each pair of 4 bars of points would be much clearer.

We believe the 3D plot in Fig 4 (previously Fig 1) is the best representation of the data – because we can show a three-way correlation between expression, sex and stimulation. 2D plots are shown in Supplementary Fig. 6C.

2. Generally, page 5-6, difficult to know exactly what is the size of effect, and key differences between cells. Giving estimated fold changes as well as direction and p value would be helpful.

We have now included effect sizes, direction, and P -values for all necessary tests.

3. Supplementary table 2 - GPR65 (2nd in table) that is coexpressed with CXorf21 is in risk locus for Crohn's, UC, MS and Ankylosing spondylitis is this of any interest/relevance?

We thank Reviewer #3 for highlighting the interesting co-expression between *CXorf21* and *GPR65*. This is an important point because there are data suggesting that this gene product operates in the lysosomal pathway, we have amended the text to state this point. We have not specifically mentioned the association with other autoimmune diseases as we are unsure of the mechanisms of action of *CXORF21* and therefore cannot provide a clear functional link between *CXORF21* and *GPR65*.

4. If 5 SNPs are "perfectly correlated" they should all have identical p values for all tests. Why do they not? And how does this enable them to focus exclusively on rs887369 before looking at H3K36me3 (ignoring that H3K36me3 does not convincingly fine map any causal

variant)?

We have now amended the manuscript so that we do not claim to have fine-mapped the causal variant down to a single SNP. Instead, throughout the manuscript, we refer to the associated haplotype, which is tagged by rs887369.

Rs887369 was directly genotyped on the Illumina HumanOmni1 BeadChip chip and therefore has complete data across the 10,995 individuals (100% genotyping rate). The four other SNPs on the associated haplotype were imputed and thus have incomplete data. This is reflected in the differences in *P* values – SNPs with missing data in some samples will have higher *P* values.

5. *Line 208 - H3K36Me3 does not get “bound” - this is a covalent histone modification modulated by for example SET2.*

Thank you – we have amended this to be clearer throughout the text.

6. *L219 It is good scientific practice to credit those who have created data/resources you then use. Please give url/reference for “ChIP resource” and the data it presents.*

We thank Reviewer #3 for spotting the omission of the reference in the main text of the manuscript. Although the reference to Schofield *et al.* (2018) for the ChIP resource was present in the methods, we have now included the citation in the results section of the manuscript (see reference 21).

7. *Line 173 - Text n=412 plot says 414 - typo?*

Thank you for noticing this error, this has been corrected to n=412 in the plot (now Fig. 2C).

8. *Figure 7 should TTS be TSS ? (typo?)*

Thank you for noticing this error, TTS has now been corrected to TSS (now Fig. 8).

Reviewer #1 (Remarks to the Author):

Author sincerely answered most of comments and revised manuscript has much improved.
Although comment #3 was not fulfilled, all other answers are generally acceptable.

Reviewer #2 (Remarks to the Author):

While the authors have not addressed all of my concerns, I am satisfied that they have done a comprehensive job of addressing the major concerns that were in the scope of this manuscript. I look forward to future manuscripts that address other queries.

Reviewer #3 (Remarks to the Author):

I have read the revised version. I am happy with all the changes.